# VolcanoFinder: Genomic scans for adaptive introgression

**Derek Setter**[1,2,3☯], **Sylvain Mousset**[1☯], **Xiaoheng Cheng**[4], **Rasmus Nielsen**[5], **Michael DeGiorgio**[6‡], **Joachim Hermisson**[1,2,7‡]

**1** Department of Mathematics, University of Vienna, Vienna, Austria, **2** Vienna Graduate School of Population Genetics, Vienna, Austria, **3** School of Biological Sciences, University of Edinburgh, Edinburgh, United Kingdom, **4** Huck Institutes of the Life Sciences, Pennsylvania State University, University Park, Pennsylvania, United States of America, **5** Departments of Integrative Biology and Statistics, University of California, Berkeley, Berkeley, California, USA, **6** Department of Computer and Electrical Engineering and Computer Science, Florida Atlantic University, Boca Raton, Florida, USA, **7** Max F. Perutz Laboratories, University of Vienna, Vienna, Austria

☯ These authors contributed equally to this work.
‡ These authors also contributed equally to this work.
\* mdegiorg@fau.edu (MD); joachim.hermisson@univie.ac.at (JH)

**Data Availability Statement:** All data are available at https://doi.org/10.5061/dryad.7h44j0zr7.

**Funding:** This research was funded by National Institutes of Health grant R35-GM128590, by National Science Foundation grants DEB-1753489,

## Abstract

Recent research shows that introgression between closely-related species is an important source of adaptive alleles for a wide range of taxa. Typically, detection of adaptive introgression from genomic data relies on comparative analyses that require sequence data from both the recipient and the donor species. However, in many cases, the donor is unknown or the data is not currently available. Here, we introduce a genome-scan method—`VolcanoFinder`—to detect recent events of adaptive introgression using polymorphism data from the recipient species only. `VolcanoFinder` detects adaptive introgression sweeps from the pattern of excess intermediate-frequency polymorphism they produce in the flanking region of the genome, a pattern which appears as a volcano-shape in pairwise genetic diversity. Using coalescent theory, we derive analytical predictions for these patterns. Based on these results, we develop a composite-likelihood test to detect signatures of adaptive introgression relative to the genomic background. Simulation results show that `VolcanoFinder` has high statistical power to detect these signatures, even for older sweeps and for soft sweeps initiated by multiple migrant haplotypes. Finally, we implement `VolcanoFinder` to detect archaic introgression in European and sub-Saharan African human populations, and uncovered interesting candidates in both populations, such as *TSHR* in Europeans and *TCHH-RPTN* in Africans. We discuss their biological implications and provide guidelines for identifying and circumventing artifactual signals during empirical applications of `VolcanoFinder`.

## Author summary

The process by which beneficial alleles are introduced into a species from a closely-related species is termed adaptive introgression. We present an analytically-tractable model for

DEB-1949268, and BCS-2001063, and by the Alfred P. Sloan Foundation. DS was funded by the Austrian Science Fund (FWF): DK W-1225-B20, Vienna Graduate School of Population Genetics (https://www.popgen-vienna.at/, https://fwf.ac.at). The funders had no role in study design, data collection and analysis, decision to publish, or preparation of the manuscript.

**Competing interests:** The authors have declared that no competing interests exist.

the effects of adaptive introgression on non-adaptive genetic variation in the genomic region surrounding the beneficial allele. The result we describe is a characteristic volcano-shaped pattern of increased variability that arises around the positively-selected site, and we introduce an open-source method `VolcanoFinder` to detect this signal in genomic data. Importantly, `VolcanoFinder` is a population-genetic likelihood-based approach, rather than a comparative-genomic approach, and can therefore probe genomic variation data from a single population for footprints of adaptive introgression, even from *a priori* unknown and possibly extinct donor species.

## Introduction

While classic species concepts imply genetic isolation [1], research of the past 30 years shows that hybridization between closely related species (or diverged subspecies) is widespread [2]. For adaptation research, this offers the intriguing perspective of an exchange of key adaptations between related species, with potentially important implications for our view of the adaptive process. Indeed, recent studies have brought clear evidence of cross-species introgression of advantageous alleles [3–6]. Well-documented examples cover a wide range of taxa, including the transfer of wing-pattern mimicry genes in *Heliconius* butterflies [7], herbivore resistance and abiotic tolerance genes in wild sunflowers [8, 9], pesticide resistance in mice [10] and mosquitoes [11], and new mating and vegetative incompatibility types in an invasive fungus [12]. Such adaptive introgressions also occurred in modern humans [13–15]: local adaptation to hypoxia at high-altitude was shown to be associated with selection for a Denisovan-related haplotype at the *EPAS1* hypoxia pathway gene in Tibetan populations [16]; positive selection has been characterized for three archaic haplotypes, independently introgressed from Denisovans or Neanderthals in a cluster of genes involved in the innate immune response [17], and immunity related genes show evidence of selection for Neanderthal and Denisovan haplotypes [18, 19].

In all examples above, evidence of adaptive introgression rests on a comparative analysis of DNA from both donor and recipient species. In particular, studies in humans often rely on maps of introgressed Neanderthal or Denisovan fragments in the modern human genome [20–22]. The tell-tale signature of adaptive introgression is a segment of mutations from the donor population that is present in strong LD and in high frequency in the recipient population [13, 16]. Unfortunately, good data from a potential donor species may not always be available, especially in the case of an extinct donor. In the absence of a donor, introgression can sometimes be inferred from haplotype statistics in the recipient species [23, 24], the most recent methods making use of machine learning algorithms based on several statistics [25]. However, as observed in [13], there is currently no framework for a joint inference of admixture and selection, such as adaptive introgression, and selection is usually inferred from the unexpectedly high frequency of introgressed haplotypes [13, 19–22, 24, 26]. A recent article [27] on adaptive introgression in plants identified four different types of studies in this field, focusing on (i) introgression, (ii) genomic signatures of selection, (iii) adaptively relevant phenotypic variation, and (iv) fitness. Our work aims to bridge the gap between classes (i) and (ii), and detect the specific genomic signature of an introgression sweep.

The genomic signature of adaptation from a *de novo* beneficial mutation has been extensively studied. When such an allele fixes in the population, the neutral alleles initially physically linked to it hitchhike to high frequency, whereas those that are initially not linked to it might be rescued from extinction by recombination, creating a valley in heterozygosity around the

selected allele—the classical pattern of a hard selective sweep [28–30]. If selection acts on standing genetic variation [31, 32], or if beneficial alleles enter the population recurrently through mutation or migration [33, 34], then multiple haplotypes from the ancestral population may survive the sweep, leading to distinctive patterns of *soft sweeps* [35, 36] with a more shallow sweep valley and typically a much weaker footprint. In the same way that recurrent migration leads to soft sweeps from *de novo* beneficial mutations, recurrent hybridization during admixture events may result in soft sweeps of adaptive introgression alleles.

In structured populations, theoretical studies have mostly focused on local adaptation and the effect of hitchhiking on differentiation indices [37–40]. However, a particularly relevant result in the context of adaptive introgression involves a structured population model with two demes connected by low migration [38]. As observed there, the pattern of a classical selective sweep is only reproduced in the subpopulation where the selected allele first arises, whereas it is highly different in the second subpopulation where the adaptive allele is later introduced by a migration event. In the latter population, heterozygosity is also reduced around the focal site, but this valley is surrounded by regions of increased heterozygosity, in which allelic variants from both subpopulations persist at intermediate frequencies.

Statistical methods to detect selective sweeps make use of patterns in both diversity within populations and differentiation among populations [41, 42]. Several widely-used tests require comparative data from two or more populations. Tests like `XP-CLR` [43] and `hapFLK` [44] can detect even soft sweeps under simple population structure and low migration rates [45]. Another family of model-based genome-scan methods identifies the effects of selection from the site frequency spectrum (SFS) and requires data from only a single population (and potentially an outgroup sequence). Using the composite-likelihood scheme suggested in [46], the `SweepFinder` software [47] detects local effects of positive selection on the SFS relative to the genome-wide genetic background SFS. The method was later extended to detect long term balancing selection [48, `BALLET`] and improved to include fixed differences in addition to polymorphic sites [49, `SweepFinder2`]. These methods compare how well two models fit the local SFS: a null model that assumes a genome-wide homogeneous SFS, and an alternative model that assumes selection acts at the focal locus. High detection power relies on modelling the specific effect of selection on the SFS for the alternative model (test 2 vs. test 1 in [47] and [48]).

The footprint of adaptive introgression, like sweeps from migration [38], differs strongly from the classical pattern of both hard or soft sweeps. The signal of adaptive introgression may therefore remain undetected by classical methods. Moreover, we are interested in distinguishing cases of adaptive introgression from adaptation within a species. For these reasons, we developed `VolcanoFinder`, a specialized method capable of detecting adaptive introgression when data from only the recipient species is available. The software and user manual are available at http://degiorgiogroup.fau.edu/vf.html.

The article is organized as follows. As a first step, we use a coalescent approach to model a recent introgression sweep in the recipient population after secondary contact with a possibly-unknown donor species. We use these results to characterize the introgression footprint by two parameters, one measuring the selection strength and the other, divergence to the donor. In the second step, these parameters are included in an extended composite-likelihood scheme, built on `SweepFinder2` [50]. We use simulated data to assess the power of our method and compare it to that of `SweepFinder2` and `BALLET`. Finally, we apply `VolcanoFinder` to human data sets in order to detect introgression sweeps in both the ancestral African and Central European populations, and we identify and discuss several candidate regions for each.

## Results

### Model and analysis

**Evolutionary history.** We consider a model with three species named *recipient*, *donor*, and *outgroup*, and their common ancestor species (see Fig 1). We assume a diploid population size $N$ for the recipient and the common ancestor, and size $N'$ for the donor. All species evolve according to a Wright-Fisher model. The recipient and the donor species diverged at time $T_d$ before present, their ancestor and the outgroup diverged at time $T_{sp} \geq T_d$. All times are measured pastward from the time of sampling in units of $4N$ generations. We assume an infinite sites model and complete lineage sorting in the ancestor, i.e. coalesence in the ancestral population occurs only between one lineage from the outgroup and one lineage from the recipient (or donor) species. Polymorphic sites in the recipient species are polarized, *e.g.*, with the help of the outgroup. If a second, more distant outgroup is available, then we also assume that fixed

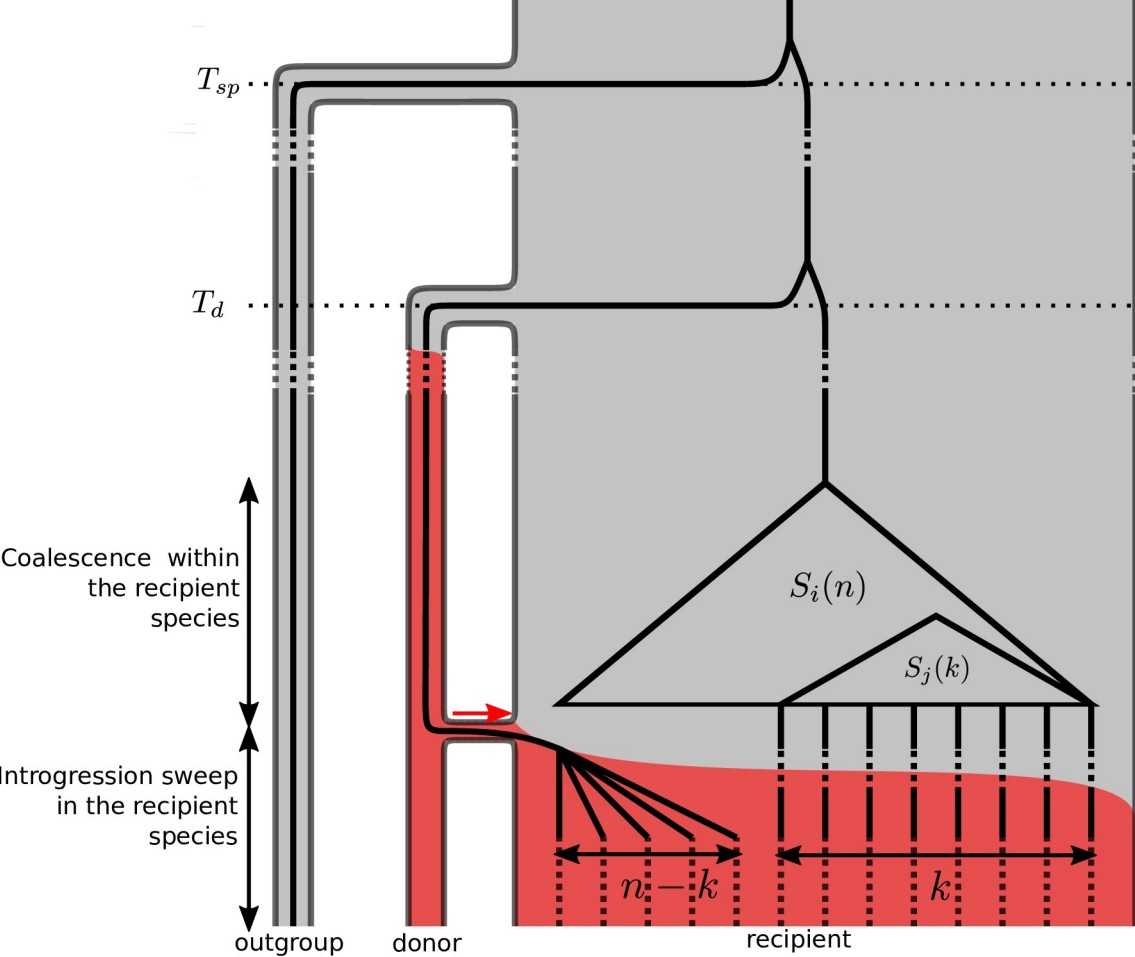

**Fig 1. Model of an introgression sweep after a secondary contact.** *Species trees*: phylogenetic relationships between the recipient, donor and outgroup species. Note that the time scale is not respected ($T_d$ and $T_{sp}$ are very large) and that all species are assumed to have the same size. *Coloured background*: frequency of the selected allele in the different species. A single favourable haplotype is introduced into the recipient species through a rare hybridization event with the donor (red arrow) where it eventually reaches fixation. *Superimposed coalescent tree*: coalescent tree of a sample of $n$ lineages, taken from the recipient population at a neutral site located at a distance $d$ from the focus of selection. $k$ lineages escape the selective sweep (see Eq (2)) and their polymorphism is a subsample of the neutral site frequency spectrum (see Eq (6)). The other $n - k$ lineages trace back as a single lineage into the donor species.

differences between the recipient species and the first outgroup are polarized. With a mutation rate (per nucleotide and generation) of $\mu$, and $\theta = 4N\mu$, the expected divergence between the recipient and the donor species is $D = 2\left(T_d + \frac{1}{2}\right)\theta$, and the expected divergence between the recipient species and its most recent common ancestor (MRCA) with the outgroup is $D_o = \left(T_{sp} + \frac{1}{2}\right)\theta$. If polarization of the fixed differences is unknown, then the full divergence $D'_o = 2D_o$ between the recipient and the outgroup species may be used instead. At time $T_i \ll T_d$, the donor and recipient species came into secondary contact, allowing for a single bout of introgression from the donor into the recipient.

Selection acts on a single locus with two alleles $B$ (derived) and $b$ (ancestral). The $B$ allele is beneficial with selection coefficient $s > 0$ for $Bb$ heterozygotes and $2s$ for $BB$ homozygotes. We assume that, prior to introgression, the $B$ allele is fixed in the donor population, but the ancestral $b$ allele is fixed in the recipient. After introgression, the $B$ allele survives stochastic loss and rises to fixation in the recipient species, sweeping away local genetic variation and pulling in foreign genetic variation in its wake. A sample of $n$ lineages from the recipient population and one lineage from the distant outgroup is sampled at the time of observation, after the fixation of the beneficial allele. We model the effect of this recent introgression sweep on the polymorphism and divergence pattern at a neighbouring neutral locus, at distance $d$ from the selected allele.

**Structured coalescent approximation.** We implemented the full model using both individual-based and coalescent-based simulations (see Materials and methods). In order to describe the key features of the selection footprint to be included into a likelihood ratio test, we used a simple analytical model based on a structured coalescent approach. The genealogy at the focal neutral locus of a sample taken from the recipient population is structured by both selection and demography. Backward in time, the coalescence process is first structured by the effects of positive selection, where we distinguish lineages that are associated with alleles $b$ and $B$ at the selected locus, like in a classical sweep model. At the time of introgression, all $B$ lineages move to the donor population, while all $b$ lineages stay in the recipient population. The further history then follows a demographic model of divergence without migration. This separation into a brief period of positive selection and a long demographic phase allows for an efficient approximation.

For simplicity, we assume in the analytical model that the sweep is initiated by a single donor haplotype. Equivalently, we can assume that all $B$ lineages quickly coalesce in the donor population (due to a bottleneck or recent origin of the $B$ allele). As a consequence, we only need to follow a single ancestral lineage in the donor population and the donor population size does not enter the results.

*Star-like approximation.* During the selective phase, the $B$ allele sweeps through the population following a frequency trajectory $X[t]$. At a neutral locus linked to the the selected site, any pair of lineages currently associated with the $B$ allele may coalesce at rate $\frac{1}{2NX[t]}$, while any single such lineage may recombine to the $b$ background at rate $R(1 - X[t])$ per generation [51]. Here, $R$ is the rate of recombination between the selected and neutral site, i.e. $R = rd$, where $r$ is the per-site recombination rate and $d$ is the distance in base pairs. Generally, $X[t]$ is a stochastic trajectory, but in large populations and for strong selection it is well approximated by a deterministic curve following logistic growth, $\dot{x}(t) = 4Nsx(t)(1 - x(t))$, where $x(0) = 1/(2N)$. In this case, any lineage at distance $d$ from the selected locus may escape the selective sweep by recombining to the $b$ background with the probability [47, 51]

$$P_e = 1 - e^{-\alpha d}, \tag{1}$$

with $\alpha = \frac{r}{s}\ln(2N)$. For strong selection, lineages recombine independently to the $b$

background, so that the probability that exactly $k$ lineages among $n$ escape the sweep is given by the binomial distribution [47]:

$$P_e(k|\alpha, d) = \binom{n}{k} P_e^k (1 - P_e)^{n-k}. \tag{2}$$

The $n - k$ lineages that do not escape the sweep coalesce instantaneously to the single ancestral lineage on which the beneficial $B$ allele first appeared. This star-like assumption ignores coalescence in the $B$ background followed by recombination into the $b$ background, but it permits an analytical approximation for the genealogical effects of the sweep even for large sample sizes $n$.

*Demographic phase.* Prior to the introgression event, coalescence of the remaining $k + 1$ lineages is structured demographically. The $k$ lineages coalesce neutrally in the recipient population, while coalescence with the single ancestral $B$ lineage only occurs once the lineages have traced back to the common ancestral population. For our analytical analysis, we make the simplifying assumption that the neutral coalescence of the $k$ escaped lineages occurs before finding a common ancestor with lineage tracing through the donor population. That is, we assume complete lineage sorting. Note that this assumption does not affect predictions of genetic diversity, which rely on a sample of $n = 2$ individuals.

## Volcanoes of diversity

The differences between introgression sweeps and classical sweeps can be seen in their respective footprint on the expected heterozygosity (pairwise nucleotide diversity) $H$ at neighboring loci. As shown in Fig 2A, introgression from a diverged donor population changes the typical valley shape of a classical sweep to a volcano shape, where diversity exceeds the genomic background in the flanking regions. We can understand this difference as follows.

Starting with a sample of size $n = 2$ taken from the recipient population directly after fixation of the $B$ allele, there are four potential coalescent histories during the sweep phase. If both lineages do not coalesce during the sweep, then one or both must have escaped the sweep by recombination. We denote the probability of these events by $P_{Bb}$ and $P_{bb}$. Alternatively, if the lineages coalesce, then their ancestral lineage can be associated with the $B$ or the $b$ allele, with respective probability $P_B$ and $P_b$. Because the star-like approximation assumes that coalescence only occurs among $B$ lineages at the start of the sweep, we have $P_b = 0$. The other probabilities are summarized in Table 1. The expected heterozygosity follows as $H = 2\theta \, \mathbb{E}[T_{coal,2}]$, with $\theta = 4N\mu$ and $\mathbb{E}[T_{coal,2}]$ the expected pairwise coalescence time, averaged over the four scenarios. Neglecting the time during the sweep, the coalescence times are entirely due to the demographic phase. For a classical sweep, this is just the neutral coalescence time in the study population (*i.e.*, $\mathbb{E}[T_{coal,2}] = 1/2$ in units of $4N$ generations, assuming standard neutrality). In the case of an introgression sweep, however, this time is increased if a single line has escaped the sweep (probability $P_{Bb}$). In this case, coalescence is only possible in the common ancestor of the donor and recipient species and $\mathbb{E}[T_{coal,2}] = T_d + 1/2$. The expected coalescence times for all cases are shown in Table 1.

Under the star-like approximation, we then obtain:

$$\begin{aligned} H_{\text{classic}} &= (1 - P_B)\theta = (P_{bb} + P_{Bb})\theta \\ H_{\text{intro}} &= (1 - P_B)\theta + 2T_d P_{Bb}\theta = P_{bb}\theta + P_{Bb}D, \end{aligned} \tag{3}$$

using $P_B + P_{Bb} + P_{bb} = 1$ and $D = (2T_d + 1)\theta$. For both introgression and classic sweeps, coalescence during the sweep ($P_B$) reduces genetic diversity, while *partial* escape through

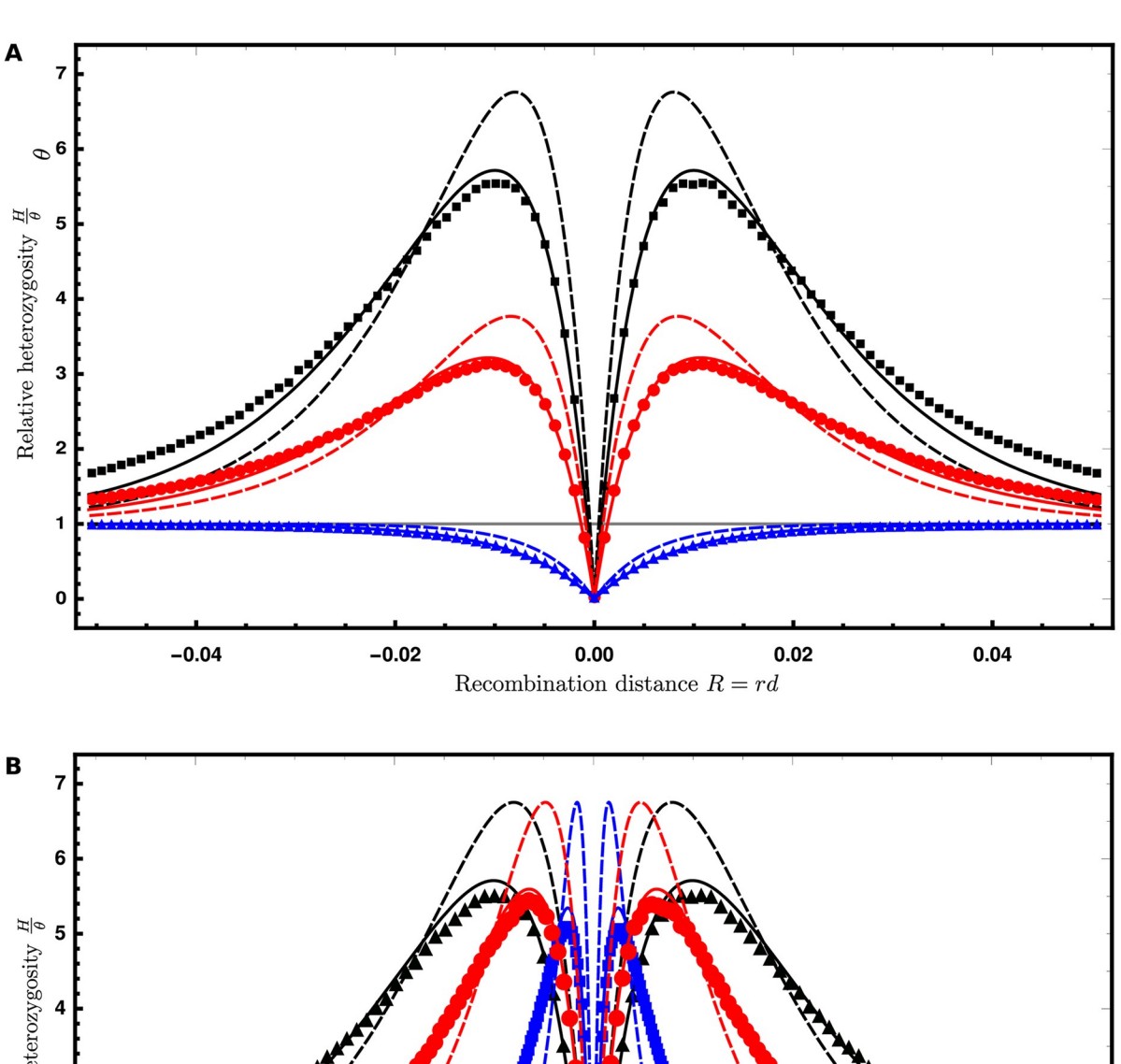

**Fig 2. Volcanoes of diversity.** Expected genetic diversity after the sweep relative to the initial heterozygosity, $\frac{H}{\theta}$ for a beneficial mutation centered at 0 as a function of the recombination distance $R = rd$ to a neutral locus on left (−) and right (+) sides. For both panels, the lines show the predictions under the star-like approximation (dashed) and the better approximation (solid, see S1 Appendix, Section 1). The dots show the average ± 3 standard errors about the mean (the error bars are smaller than the plot points). **A**. The effect of divergence of the donor population for an introgression sweep with $2Ns = 1000$. The divergence time (in units of $4N$ generations) is $T_d = 6$ (*i.e.*, $D = 13\theta$, black), 3 ($D = 7\theta$, red), and 0 ($D = \theta$, blue), where $T_d = 0$ is a classic sweep from a *de novo* mutation. **B**. The effect of the strength of selection for an introgression sweep with $T_d = 6$ ($D = 13\theta$). The strength of selection is $2Ns = 1000$ (black), 600 (red), or 200 (blue). For both panels, $\theta = 0.002$ ($N = 5000$, $\mu = 10^{-7}$), $r = 10^{-7}$, and the window size is 100 nt.

**Table 1. Summary of the effects of selection.**

| Genealogical history | Probability star-like approx. | Coalescence times | |
|---|---|---|---|
| | | classic sweep | introgression sweep |
| $\{B, B\} \rightarrow \{B\}$ | $P_B = e^{-2\alpha d}$ | 0 | 0 |
| $\{B, B\} \rightarrow \{B, b\}$ | $P_{Bb} = 2(1 - e^{-\alpha d})\, e^{-\alpha d}$ | 1/2 | $T_d + 1/2$ |
| $\{B, B\} \rightarrow \{b, b\}$ | $P_{bb} = (1 - e^{-\alpha d})^2$ | 1/2 | 1/2 |
| $\{B, B\} \rightarrow \{b\}$ | $P_b = 0$ | 0 | 0 |

Possible (backward) coalescence or recombination events during the selective sweep with $n = 2$ lines linked to the beneficial allele at the time of sampling $\{B, B\}$, probabilities under the star-like approximation, and expected time to coalescence for both a classic sweep and an introgression sweep. Note that all times are measured in units of $4N$ generations.

recombination ($P_{Bb}$) increases diversity only in the introgression case. Substituting the probabilities from Table 1, we obtain the expected heterozygosities as functions of $\alpha d = (R/s)\ln(2N)$ and $D$. In Fig 2 (dashed lines) they are shown together with simulation data as function of the recombination distance $R$ and $D$ and $s$ as parameters. Fig 2A shows the effect of the divergence $D$ of the donor population and Fig 2B shows the effect of the strength of selection $s$ acting on the the beneficial allele. While divergence mostly affects the height of the volcano for introgression sweeps, the selection strength mostly scales the width of the footprint.

We can analyze the shape of the footprint in more detail using the star-like approximation. In this case, the width of the signal can be measured in terms of a single compound parameter $\alpha d = R \log(2N)/s$. This compound parameter is a generalized description of the effect of a sweep along the genome, as distance from the sweep center is measured relative to the strength of selection. The top panel of Fig. A5 in S1 Appendix shows the effect of the adaptive introgression sweep on genetic diversity as a function of $\alpha d$. When $\alpha d$ is near 0, diversity is reduced relative to the background, while at distance $\alpha d \approx 1$, we see the peak of the volcano pattern. At distances $\alpha d \approx 6$, the sweep has a much smaller effect and diversity is only slightly higher than the genomic background. We describe this formally in what follows.

For a classical hard selective sweep, we find that the variation at a scaled distance $\alpha d = \frac{1}{2}\ln(1/X)$ from the selected site is reduced by a fraction of $X$ (i.e., $H_{\text{classic}} = (1 - X)\theta$). Due to the excess variation that is brought in from the diverged donor population, the central valley of an introgression sweep is narrower, with decreasing width as divergence $D$ increases. At a distance

$$\alpha d = \ln\left(\frac{2D - \theta}{2D - 2\theta}\right) \underset{D \to \infty}{\longrightarrow} 0 \tag{4}$$

both effects compensate and we obtain an expected heterozygosity of $H_{\text{intro}} = \theta$. At larger distances, $H_{\text{intro}}$ overshoots the background level and assumes a maximum value of

$$H_{\text{intro}}^* = \frac{D^2}{2D - \theta},$$

which is independent of the selection coefficient in the star-like approximation. Using $D = (2T_d + 1)\theta$, we can express the relative height of the "volcano" above the background level as a function of the divergence time

$$\frac{H_{\text{intro}}^* - \theta}{\theta} = \frac{4T_d^2}{4T_d + 1}.$$

This maximum is reached at a scaled distance $\alpha d = \ln\left(\frac{2D-\theta}{D-\theta}\right) \xrightarrow{D\to\infty} \ln(2) \approx 0.7$. The signal of the introgression sweep is therefore strongest at the distance where a classical sweep signal has already decayed by at least 75%. At a scaled distance of

$$\alpha d = \ln\left(\frac{2D-\theta}{D-\theta}\right) + \ln\left(\frac{10}{10-3\sqrt{10}}\right) \xrightarrow{D\to\infty} \ln\left(\frac{20}{10-3\sqrt{10}}\right) \approx 3.7 \qquad (5)$$

the increased heterozygosity returns to 10% of the maximum value, and $H_{\text{intro}} = \theta + \frac{H^*_{\text{intro}} - \theta}{10}$. The footprint of an introgression sweep is therefore considerably wider than that of a classic sweep, in which, a 90% recovery of the decreased diversity is expected at distance $\alpha d = \frac{1}{2}\ln(10) \approx 1.2$.

**Beyond the star-like approximation.** While the star-like approximation (dashed lines in Fig 2) provides qualitatively accurate results, it overestimates $P_{Bb}$, and consequently, the height of the volcano peaks. Simulations show that this height may also be slightly dependent on the selection coefficient (compare dashed lines and dots for simulated values in Fig 2B). In the supplementary information, we provide a more accurate approximation for the probabilities in Table 1 using a stochastic approach based on Yule branching processes [51]. In particular, this approach allows for coalescence during the sweep as well as recombination of coalesced lineages to the $b$ background. We thus obtain $P_b > 0$ and reduced values for $P_{Bb}$ relative to the star-like approximation. As shown in Fig 2 (solid lines, see also S1 Appendix, Section 1), this leads to an improved fit of the simulation data for pairwise diversity. However, an extension of this method to the site-frequency spectrum for larger samples is difficult. We therefore resort to the star-like approximation in what follows and in our parametric test.

**Single iterations.** Footprints of introgression sweeps, like classical sweeps [46], are highly variable due to the stochastic events in the genealogical history of the sampled lineages. Single numerical replicates, as well as patterns in data, can deviate strongly from the "expected" volcano shape displayed in Fig 2. In Fig 3, we show a typical set of introgression footprints obtained from single replicate runs (see Fig. A2 in S1 Appendix for more examples). We see that, under favorable conditions (large $T_d$ and sampling directly after the fixation of the beneficial allele), volcano shapes are clearly discernible even in single iterations. However, we also see that the width and symmetry of the volcanoes varies greatly between replicates. The key reason for this variation is the early recombination events during the initial stochastic establishment phase of the beneficial allele. In the sample genealogy, the B-linked allele can dissociate from the foreign haplotype if even a single recombination event occurs in the time between coalescence of all B-linked lineages and the initial introgression of the B allele. As the volcano pattern is relatively broad, these recombination events occur with substantial probability. At distances beyond the recombination break point, only genetic variation from the recipient population hitchhikes, resulting in the classic sweep pattern from *de novo* mutation. As an example, compare the independent replicate simulations in panels *C* and *D* of Fig 3. The simulation in panel *D* resulted in the broad volcano pattern we expect based on the analytic model. In panel *C*, an early recombination to the right of the beneficial mutation prevented the hitchhiking of foreign genetic variation, and beyond this position, genetic diversity is not elevated. Since independent recombination events are required to "cut the volcano" on both sides of the beneficial allele, strong asymmetries in the shape arise naturally, and we see this asymmetry among the 4 replicate simulations shown in Fig 3.

## The footprint of adaptive introgression in the SFS

Following [47] we use a parametric approach to model the effect of a recent introgression sweep on the site frequency spectrum (SFS) at distance $d$ from a recently-fixed beneficial allele.

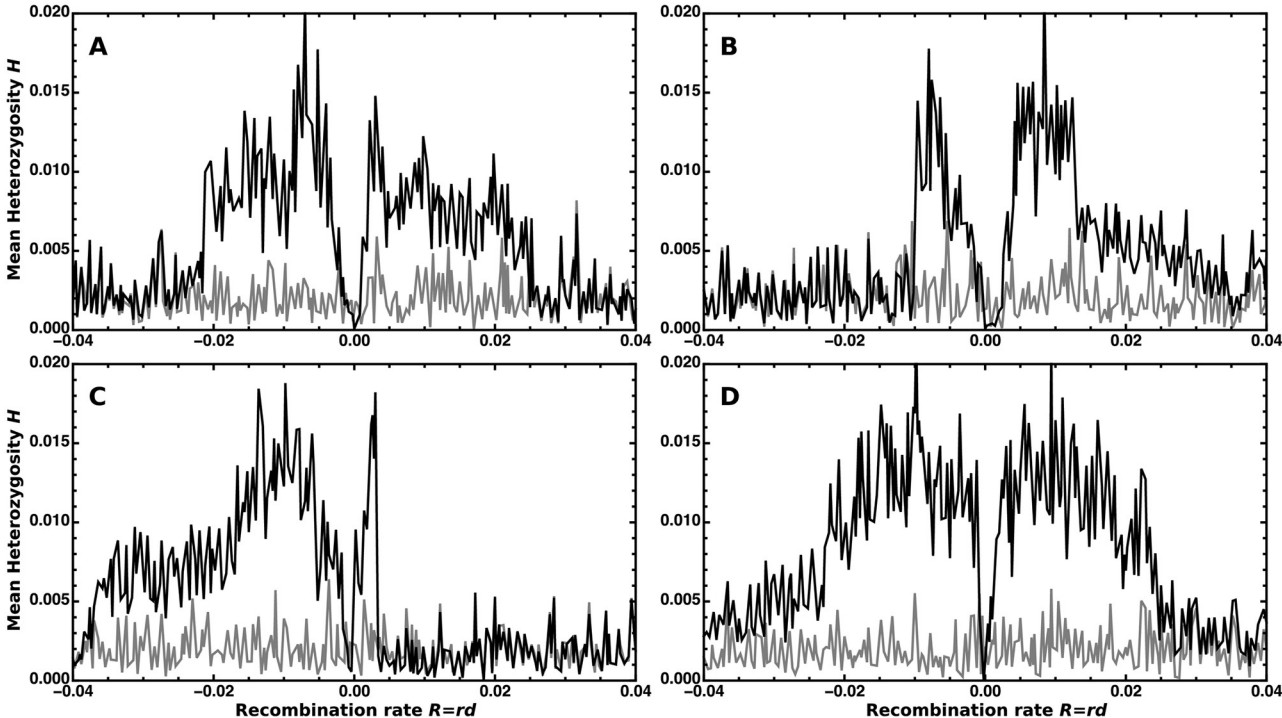

**Fig 3. Single iterations of an adaptive introgression event.** Each of the panels A, B, C, and D shows an independent, randomly chosen simulation run. We calculated the whole-population mean genetic diversity in 401 non-overlapping non-adjacent one kb windows separated by one kb and centred on the selected locus. The initial heterozygosity and the genetic diversity at fixation of the beneficial $B$ allele are shown in grey and black, respectively. Here, $\theta = 0.002$ ($N = 5\,000$, $\mu = 10^{-7}$), $r = 10^{-7}$, $T_d = 6$ ($D = 13\theta$), and $s = 0.06$ ($2Ns = 600$).

Our model includes the compound parameter $\alpha$ (sweep strength) of the classic hard sweep model in [47], as well as the additional parameter $D$ (donor divergence) specific to an introgression sweep.

**The background reference SFS.** Consider an alignment of $n$ sequences from the recipient species and one sequence from an outgroup species to polarize the data. In the recipient population, we observe a mutation with frequency $i = 1, 2, \ldots, n$ with probability $S_i(n)$, where $S_n(n)$ is the probability of observing a fixed difference relative to the outgroup. The $S_i(n)$ represent the non-normalized SFS, i.e. the probability of a monomorphic site is $1 - \sum_{i=1}^{n} S_i(n)$. If we sample a second more distant outgroup, under the assumptions of complete lineage sorting and the infinite sites mutation model, we can further distinguish the lineage on which the fixed differences occur. In this case, $S_n(n)$ is the probability that the mutation occurred specifically on the lineage ancestral to the recipient population, and we denote by $S_0(n)$ the per-site probability of observing a mutation private to the first outgroup lineage. That is, the probability of observing a fixed difference is $S_0(n) + S_n(n)$. If a second outgroup is unavailable, then only polymorphic mutations in the recipient species can be polarized, but not the fixed differences. In this case, we arbitrarily label the state in the first outgroup as "ancestral" such that $S_0(n) = 0$. Following [47], the neutral reference SFS can be estimated from the observed genome-wide data. Given these estimates for the $S_i(n)$, the spectral probabilities $S_j(k)$ in subsamples of $k \leqslant n$ sequences follow as

$$S_j(k) = \sum_{i=j}^{n} S_i(n) \frac{\binom{i}{j}\binom{n-i}{k-j}}{\binom{n}{k}}. \tag{6}$$

The conditional probability of observing $i$ mutant alleles among $n$ lineages given that the site is polymorphic is

$$p_{i,n} = \frac{S_i(n)}{\sum_{j=1}^{n-1} S_j(n)}. \tag{7}$$

Similarly, the conditional probability of observing $i$ mutant alleles given that the site is polymorphic or a fixed difference for which the recipient species has the inferred derived state is

$$q_{i,n} = \frac{S_i(n)}{\sum_{j=1}^{n} S_j(n)}. \tag{8}$$

If the mutation rate $\mu$ varies along the genome, then the probabilities $S_i(n)$ will vary among sites, because $S_i(n)$ is proportional to $\theta = 4N\mu$. In contrast, the mutation rate cancels in the conditional probabilities $p_{i,n}$ and $q_{i,n}$, which are expected to be constant along the genome.

**The expected SFS after the sweep.** Following the star-like approximation, $k$ lineages escape the introgression sweep with probability $P_e(k|\alpha, D)$ (Eq 2). We assume complete lineage sorting between these lineages and the single ancestral lineage of all lines that are caught in the sweep and transition to the donor species. An introgression sweep then transforms the SFS as follows. Let $S_i'(n|\alpha, d, D)$ denote the per-nucleotide probability of observing $i$ mutant lineages in a sample of $n$ lineages from the recipient species after an introgression sweep with strength parameter $\alpha$ and divergence parameter $D$ at distance $d$. Below, we assume that the time for the coalescent process in the recipient species is negligible relative to the divergence time between the recipient and the donor species (see Fig 1). As shown in the supplement (see S1 Appendix, Section 2), this assumption can be relaxed. However, because the more complex model did not lead to a clear improvement of our statistical test, we focus on the simple approximation in the main text. In this case, the transformed SFS after the introgression sweep is given by ($1 \leqslant i \leqslant n-1$):

$$S_i'(n|\alpha, d, D) = \left( \sum_{k=i+1}^{n} P_e(k|\alpha, d) S_i(k) \right) + P_e(n-i|\alpha, d)\frac{D}{2} + P_e(i|\alpha, d)\frac{D}{2}. \tag{9}$$

The first term on the right hand side accounts for the contribution of mutations that occur during the coalescent process of the escaping lineages in the recipient species. The second and third terms, respectively, account for mutations on the long ancestral lineages in the donor and recipient population, which partition the $n - i$ lineages that are caught in the sweep from the $i$ escape lineages. Because the expected coalescence time for these lineages is $T_d + 1/2$, the probability for a mutation to hit either lineage is $\theta(T_d + 1/2) = D/2$. The conditional probabilities given that the site is polymorphic follow as

$$p_{i,n}'(\alpha, d, D) = \frac{S_i'(n|\alpha, d, D)}{\sum_{j=1}^{n-1} S_j'(n|\alpha, d, D)}. \tag{10}$$

Because all terms in Eq (9) are proportional to $\theta$, this normalization removes the dependence on the mutation rate, analogous to Eq (7).

If fixed differences are polarized, then a site will be a fixed difference for which all recipient lineages carry the mutant allele if a mutation occurred in the lineage that connects the MRCA of the sample to the MRCA of the recipient and the outgroup species, leading to the following

probability:

$$S'_n(n|\alpha, d, D) = \left( (D_o - \tfrac{D}{2}) \sum_{k=1}^{n-1} P_e(k|\alpha, d) \right) + D_o P_e(0|\alpha, d) + D_o P_e(n|\alpha, d), \qquad (11)$$

where $D_o$ is the expected divergence between the recipient species and the MRCA of the recipient and the outgroup species. The first term in the right-hand side of Eq (11) accounts for the cases when some (but not all) lineages escape the sweep, whereas the second and third terms account for the cases when no lineages or all lineages escape the sweep. In our secondary contact model, $D_o$ can be estimated from the data as

$$D_o = S_n(n) + \frac{1}{n} \sum_{i=1}^{n-1} i S_i(n), \qquad (12)$$

which is equivalent to $D_o = S_1(1)$, as can be seen from Eq (6). The second term on the right hand side of Eq (12) is the mean number of mutations accumulated in each recipient lineage since their MRCA, related to the unbiased estimator of $\theta$, $\hat{\theta}_L = \frac{1}{n-1} \sum_{i=1}^{n-1} i S_i(n)$ [52, eq. (6) and (8)]. If fixed differences are not polarized, then Eqs (11) and (12) still hold when substituting $D_o$ with the full divergence between the recipient species and the outgroup $D'_o$. Assuming constant mutation rates between the focal species and the outgroup, all three terms in Eq (11) are proportional to $\theta$, making the conditional probabilities once again independent of the mutation rate,

$$q'_{i,n}(\alpha, d, D) = \frac{S'_i(n|\alpha, d, D)}{\sum_{j=1}^{n} S'_j(n|\alpha, d, D)}. \qquad (13)$$

**A composite likelihood ratio test.** Our test builds on the composite-likelihood method first introduced in [46] and further developed in [47–49]. Sequence data are collected in an alignment of $n$ chromosomes from the recipient species and possibly one chromosome from an outgroup species. We assume that mutations are polarized and consider only informative sites, *i.e.*, sites for which at least one chromosome in the recipient species harbors the inferred derived allele. Let $L$ be the number of informative sites and $X_\ell$ the frequency of the derived allele at the $\ell$th informative site. We contrast the composite likelihoods of a reference and an alternative model for the empirical SFS. The reference model assumes that the distribution of the classes in the SFS is homogeneous along the chromosome. Accounting for fixed differences, the genome-wide SFS conditional probabilities are given by Eq (8), and the composite likelihood of the reference model is

$$CL_0 = \prod_{\ell=1}^{L} q_{X_\ell, n}. \qquad (14)$$

The alternative model assumes that an introgression sweep event with unknown parameters $\alpha$ and $D$ recently happened at some location on the chromosome, leading to an inhomogeneous altered SFS along the chromosome. Let $d_\ell$ be the distance of the locus of the introgression sweep to the $\ell$th informative site. The composite likelihood $CL_1$ of the alternative model including fixed differences uses the local SFS conditional probabilities from Eq (13),

$$CL_1(\alpha, D) = \prod_{\ell=1}^{L} q'_{X_\ell, n}(\alpha, d_\ell, D). \qquad (15)$$

If fixed differences with a single outgroup are unavailable (for instance if different outgroup species were used to polarize polymorphic sites), then the test can also be set up without fixed differences, by using probabilities $p_{X_\ell,n}$ from Eq (7) in Eq (14) and $p'_{X_\ell,n}(\alpha, d_\ell, D)$ from Eq (10) in Eq (15).

For a given genomic position of the beneficial allele, maximum composite likelihood estimates $\hat{\alpha}$ and $\hat{D}$ are obtained such that $CL_1(\hat{\alpha}, \hat{D}) = \max_{\alpha,D}(CL_1(\alpha, D))$ with $\alpha > 0$ and $0 \leqslant D \leqslant 2D_o$ if fixed differences are polarized and $0 \leqslant D \leqslant D'_o$ otherwise. The test statistic for the composite likelihood ratio test is defined as

$$T_1 = 2(\ln CL_1(\hat{\alpha}, \hat{D}) - \ln CL_0). \tag{16}$$

Although the star-like model is only a rough approximation, it allows for considerable flexibility to fit empirical patterns via optimization of the parameters $\alpha$ and $D$. While $\alpha$ modulates the width of the footprint, $D$ mostly scales the height of the volcano. As shown in Fig 3, the width of the pattern varies strongly between replicates. The model can partially compensate for this variation by adjusting $\alpha$. Still, the average estimate of $\alpha$ (across replicates) closely matches the true value in simulated data (see S1 Appendix, Section 5). In contrast, the divergence is systematically underestimated. The downward bias of $D$ compensates for the overestimation of the volcano height under the star-like approximation (Fig 2). This bias is not a problem as long as the method is only used to infer adaptive introgression, and no biological interpretation is attached to the fit parameters. Bias in $D$ needs to be accounted for, however, if the method is used for biological parameter estimation.

## Power analysis

In this section, we investigate the power of our new method VolcanoFinder to detect an adaptive introgression sweep against the genomic background signal. Typically, analyses of test power display the true positive rate of the test against the false positive rate in a so-called receiver operating characteristic (ROC) curve [45, 48, 49]. We provide ROC curves of this type in the supporting information (see Fig. B6 and Fig. B7 of S2 Appendix, Section 3). Below, however, we present an alternative analysis that is closer to the use of the test in a real genome scan. Such a scan results in a list of outlier peaks in the CLR score. It is thus natural to ask for the probability that the true positive peak ranks among the highest peaks in a larger genomic region. To do this, we must first define what counts as an independent outlier peak in the test scores of the VolcanoFinder scan. We use two approaches to identify such peaks, which are described in detail in S3 Appendix, Section 1. Briefly, one approach uses the breadth of the sweep ($\alpha$) reported by VolcanoFinder to distinguish independent peaks and to determine whether a signal is a true-positive detection of the adaptive allele. The other approach simply identifies independent maxima in the sequence of test scores along the genome.

**Local genomic region.** As a first step, we investigate the prominence of the sweep signal among the variation in the local region of the genome. Here we simulate an adaptive introgression sweep that occurs in the center of a 10 Mb (20 centiMorgan) genomic region, and we sample the population at the time of fixation of the beneficial mutation. Two scenarios are compared in this analysis. The first scenario is close to our theoretical model and describes adaptive introgression as a rare event that stems from a single successful hybridization with a highly diverged donor. In the simulations, we assume a single generation of migration with rate $m = 1/N$ from the donor to the recipient and $T_d = 4.0$. We condition on the rare case that the adaptive allele reaches fixation in the recipient population. The second scenario describes a much stronger hybridization pulse that results in fixation of the adaptive allele with high

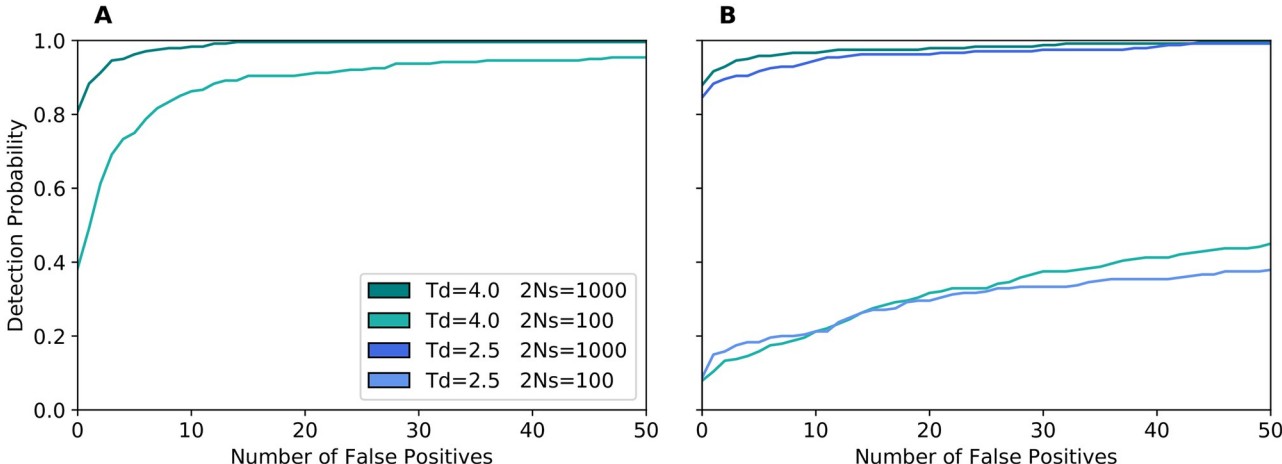

**Fig 4. Power to detect the adaptive introgression allele.** Here we plot the detection probability for `VolcanoFinder` as a function of the number of false positive signals from the genomic background that produce a higher peak. Panel A shows introgression from a rare hybridization event. Panel B shows introgression with higher migration rates: $m = 30/N$ for strong selection $2Ns = 1000$ and $m = 300/N$ for weak selection $2Ns = 100$. Divergence of the donor is $T_d = 4.0$ (green) or $T_d = 2.5$ (blue) in units of $4N$ generations. Here, $N = 10,000$ diploid individuals, mutation rate $\mu = 1.25 \times 10^{-8}$ per site per generation, and recombination rate $r = 5 \times 10^{-7}$ per site per generation.

(95%) probability (see Fig. B1 of S2 Appendix, Section 1). In this case, a single generation of migration occurs at rate $m = 30/N$ for strong selection $2Ns = 1000$ or rate $m = 300/N$ for weak selection $2Ns = 100$, and we consider a donor population with a divergence time of either $T_d = 4.0$ or $T_d = 2.5$ in units of $4N$ generations. We highlight the main results below and present a detailed analysis in S3 Appendix, Section 2.

Fig 4 shows the power to detect the adaptive introgression allele under these three scenarios: the rare hybridization scenario is shown in panel A while the high-migration scenarios are shown in panel B. In panel A, we observed that `VolcanoFinder` has very high power to detect the adaptive inrogression sweep when selection is strong. When selection is weak, the method has moderate power to detect the introgression sweep as the top peak (false positives = 0), and the power increases substantially as we considered a larger number of candidate signals. In panel B, we observed that with higher migration rates, `VolcanoFinder` retains high power to detect the sweep when selection is strong ($2Ns = 1000$, $m = 30/N$). However, we observed a substantial reduction in power when selection is weak ($2Ns = 100$, $m = 300/N$). Because the height of the volcano pattern depends on the divergence of the donor population, we expect `VolcanoFinder` to perform better with increasing levels of divergence. Here, we observed only a weak effect of divergence on the power of `VolcanoFinder` (panel B, blue vs. green lines, i.e. $T_d = 2.5$ vs. $T_d = 4.0$).

We conclude that adaptive introgression produces a highly conspicuous signal if it originates from rare hybridization with a strongly diverged donor. In contrast to a classical sweep, this even holds for weak selection: As long as the adaptive allele succeeds to establish and fix, volcano slopes are produced. Strong selection is important, however, to create a prominent local volcano signal against the background variation if the rate of neutral introgression into the genomic background is high. Since larger divergence to the donor increases both the signal and the noise, its effect on the test power partially cancels out.

**Large genomic background.** In a second step, we now assess the power of `Volcano-Finder` to detect an adaptive introgression sweep in the context of a large, contiguous chromosome or even an entire genome. We compare this power to the one `SweepFinder2` (a method that has been designed for classical sweeps) and `BALLET` (a scanner for long-term

balancing selection). We are interested in the effect of five parameters on the statistical power: the selection coefficient $s$ of the beneficial allele, the split time $T_d$ between the donor and the recipient population, the time elapsed since the end of the introgression sweep $T_s$, the presence of polymorphic genetic variation co-introgressing with the beneficial allele (hard or soft introgression sweeps, see below), and the admixture level of the reference genomic background.

Ideally, we would like to simulate genome-wide polymorphism data in single simulation runs. However, even using SLiM3 [53] and msprime [54], the most efficient methods currently available, we were still limited to 10 Mb (20 centiMorgan) genomic regions and even then, only a handful of parameter combinations. In this section, we therefore performed pure coalescent simulations using msms [55], which permits an expansive analysis of Volcano-Finder's power. While this approach is fast, it is limited in two ways. First, only much smaller genomic regions can be efficiently simulated in each single run. We therefore simulated the genomic background by proxy, using a large number of replicate 200 kb regions. We validated this approach by constructing *chimeric* chromosomes and comparing the power to that of contiguous 10 Mb chromosomes. Our results in S3 Appendix, Section 5, show that this approximation does not bias the power estimates. Second, it is impossible to condition on fixation of the adaptive allele when combining demographic history and positive selection in msms. To circumvent this limitation, we consider only the high-migration scenario described above, in which fixation of the beneficial mutation occurs anyway with high probability.

**Hard and soft introgression sweeps.** Hard and soft selective sweeps refer to sweeps that originate from a single or multiple copies of the beneficial allele, respectively. In the case of introgression, hard sweeps trace back to a single migrant from the donor population, while soft sweeps originate from multiple migrants. More generally, hard introgression sweeps represent all scenarios where the beneficial haplotype traces back to a very recent common ancestor in the donor population, such that no standing genetic variation from the donor population can co-introgress with the beneficial allele. Conversely, soft introgression sweeps allow for diversity among the introgression haplotypes. In our simulations, we maximize this diversity by assuming that the beneficial allele has fixed in the donor population a long time ago. As a consequence, all introgression haplotypes are related by a standard neutral coalescent in a donor population of size $N' = N$. While classical hard and soft sweeps in a single population can lead to strongly diverging footprints [36], hard and soft introgression sweeps both lead to very similar volcano patterns in the heterozygosity (compare Fig. B4 and Fig. B5 of S2 Appendix). The central valley is slightly deeper for hard introgression sweeps, and the peaks are slightly higher for soft introgression sweeps.

**Admixture in the genomic background.** We perform two series of power analyses, with different assumptions about admixture in the genomic background, analogous to our 10 Mb analysis. In the first scenario, we assume that secondary contact does not lead to introgression in the genomic background, but only around the selected locus. We thus test for signals of *local introgression* at the target locus against the alternative assumption of *no introgression*. The results are presented in S2 Appendix, Section 4. Analogous to the result in Fig 4A, we observe very high power to detect hard and soft introgression sweeps for strong and weak selection as long as the divergence to the donor population is sufficiently large ($T_d \geq 2.5$). For all parameter values, the specialized VolcanoFinder method is more powerful than the alternative methods. Below, we discuss in more detail our second (and more challenging) scenario, where introgression leads to genome-wide admixture. That is, we test for the power to detect *adaptive introgression* against a background of *neutral introgression*, with a uniform genome-wide admixture proportion.

Both scenarios represent limiting cases of adaptive introgression events that may be observed in nature. If introgression is a very rare event and/or introgressed variation is usually

deleterious and purged from the recipient population by selection, then a non-admixed background is the appropriate reference. Conversely, genome-wide admixture can be expected with higher admixture rates and if genetic barriers to gene flow are weak.

**Statistical power with an admixed reference genomic background.** Fig 5 and Fig. B7 of S2 Appendix show the power of all three tests assuming a constant genome-wide admixture

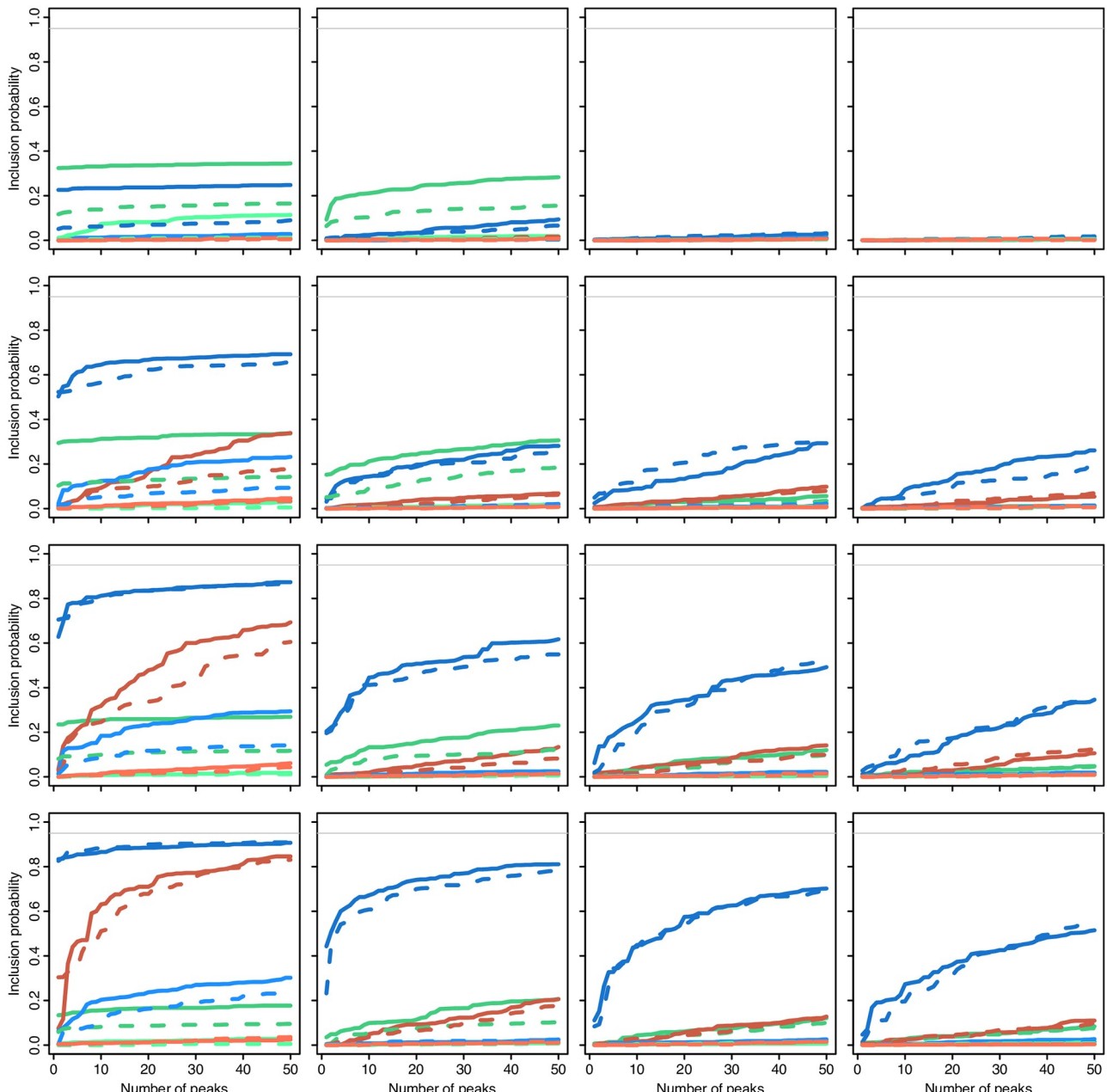

**Fig 5. Detection probability of an introgression sweep (admixed background).** Probability of an introgression sweep event to be detected in a genome-scan analysis using VolcanoFinder (blue), BALLET (brown) and SweepFinder2 (green). The x-axis in represents the number of false-positive peaks from the neutral data which score higher than the true-positive signal. The donor species diverged from the recipient species at (top to bottom) $T_d$ = 1, 2.5, 4, 5.5 (i.e. $D$ = 3$\theta$, 6$\theta$, 9$\theta$, 12$\theta$) and the selective sweep ended (from left to right) $T_s$ = 0, 0.1, 0.25, 0.5 units of 4$N$ generations in the past. Solid lines: no polymorphism in the donor species (hard introgression sweep). Dashed lines: polymorphism exists in the donor species (possible soft introgression sweep). Dark colour: 2$Ns$ = 1000; light colour: 2$Ns$ = 100. Analyses involved a neutral admixed genomic background with the same level of admixture as a reference.

proportion. Because this proportion is adjusted such that an introgression sweep occurs in 95% of all simulation, the maximal power (detection probability) that can be achieved by a "perfect" test in this case is 0.95 (as observed in the figures). It also means that the admixture proportion is larger for weak selection (3% for $2Ns = 100$) than for strong selection (0.3% for $2Ns = 1000$).

As observed for the 10 Mb simulations (Fig 4B), high levels of admixture in the genomic background leads to a strong reduction in power for all three methods (compare Fig. B3 of S2 Appendix to Fig 5). Due to the higher admixture rate, this holds, in particular, for simulations with weak selection ($2Ns = 100$) whereas the reduction is moderate for $2Ns = 1000$. All methods need a relatively high false discovery rate to achieve rejection rates close to the expected maximum (Fig. B7 of S2 Appendix), thus reducing the probability of an introgression sweep to be detected in small sets of outlying peaks. VolcanoFinder still performs better than other methods (Fig 5), especially for recent introgression sweeps ($T_s = 0$) from donor species that are not too closely related ($T_d \geqslant 2.5$, *i.e.*, $D \geqslant 6\theta$). For instance, a recent introgression sweep ($T_s = 0$) from a moderately diverged donor species ($T_d = 2.5$, $D = 6\theta$) with a strongly selected allele ($2Ns = 1000$) will be associated with the genome-wide highest CLR with probability around 1/2 for VolcanoFinder 1/3 for SweepFinder2 and close to 0 for BALLET. Notably, VolcanoFinder maintains some statistical power for much older selective events ($T_s \geqslant 0.25$, i.e., more than $N$ generations) when the detection probability of other tests is close to 0.

**Robustness of VolcanoFinder to long-term balancing selection.**   Balancing selection increases the polymorphism-to-divergence ratio in regions surrounding the selected site [56]. Because this signal also occurs in the case of an introgression sweep, VolcanoFinder could falsely detect an introgression sweep in the case of long term balancing selection. To assess the robustness of VolcanoFinder, we compared the rejection rates of VolcanoFinder and BALLET under three demographic models inspired by [48] for increasingly old balancing selection (overdominance). The results are shown in Fig. B19 of S2 Appendix. Unlike BALLET, the rejection rate of VolcanoFinder is close to the false positive rate (although a bit larger) for moderately old balanced polymorphisms ($T_s \leqslant 8.75$) and remains low (10% to 20% depending on the demographic model) even for very old balanced polymorphisms ($T_s = 20$). Interestingly, the effect of the demographic model on the power to detect the footprints of balancing selection acts in opposite directions for VolcanoFinder and BALLET, suggesting that these two methods are sensitive to opposite patterns in the SFS. Overall, VolcanoFinder was found to be relatively robust to long-term balancing selection.

**Robustness of VolcanoFinder to classic sweeps.**   In Fig 5, we observed moderate power for SweepFinder2 to detect a strong introgression sweep. In particular, when divergence is very low (Panel 1-1), this method outperforms VolcanoFinder. However, when selection is weak, SweepFinder2 has very low power. In S3 Appendix, Section 3, we simulate a classic sweep from *de novo* mutation in a panmictic population and compare the power of both methods to detect the sweep. While SweepFinder2 has high power when selection is strong and moderate power when selection is weak (Fig. C6 Panel A of S3 Appendix), VolcanoFinder achieved only moderate power when selection is strong and very low power when selection is weak (Fig. C6 Panel B). In panel C, we observed that the background scores are on average higher for VolcanoFinder, however, there is little difference in the range of high-valued outlier scores in the genomic background. Rather, the power to detect the sweep primarily depends on the strength of the true-positive test scores obtained in the center-most region of the sweep. In classic sweep scenarios, elevated test scores indicative of positive selection were observed only for a small region near the center of the sweep (Fig. C7 of S3 Appendix), which contrasts with the breadth of the signal observed for adaptive introgression sweeps (Fig. C4 and Fig. C5 of S3 Appendix). Importantly, we found that for both strong and weak

selection, the power of `SweepFinder2` to detect classic sweeps closely corresponds to the power of `VolcanoFinder` to detect adaptive introgression sweeps.

**Robustness of `VolcanoFinder` to background selection.** The reduction in genetic diversity at neutral sites through the purging of deleterious variation at nearby negatively-selected loci is known as background selection [57]. Although the effect of background selection extends only short distances in the genome, it generates a skew in the site frequency spectrum toward low-frequency alleles [58], which resembles the effect of positive selection on linked neutral variation. For a scenario of panmixia, in S3 Appendix, Section 4, we simulate the effects of ubiquitous background selection acting in the genome using realistic gene structures based on annotations in the RefSeq database and a complementary set of simulations with purely-neutral evolution. We applied `VolcanoFinder` on the two data sets and compare the test scores under neutrality to those under background selection. We observed that the distribution of scores under background selection is nearly identical to that under neutrality (Fig. C8 Panel A of S3 Appendix). Background selection did produce a few higher-valued outlier peaks relative to the scores observed under neutrality (Panel B), however, the effect is very small (Panel C) relative to the strength of true-positive test scores we observed for simulations of positive selection (both for adaptive introgression sweeps in Fig. C2 of S3 Appendix and classic sweeps from *de novo* mutation in Fig. C7 of S3 Appendix). We therefore conclude that `VolcanoFinder` is robust to the effects of ubiquitous background selection.

**Out of Africa.** Finally, we investigate the power of `VolcanoFinder` under the out-of-Africa human demographic model inferred in [59]. In this case, the donor population diverged either 615 kya (Neanderthal-like), 1.230 mya, or 2.460 mya, corresponding to $D/\theta = 2$, 3, or 6, respectively, and selection on the adaptive allele is strong ($2Ns = 1000$). We assume that the hybridization event occurred in the ancestral Eurasian population after the population expansion such that the adaptive allele reaches fixation just before the split into separate European and Asian populations, and we consider both the case where the genomic background is impermeable to introgressive variation and the case where neutral introgression occurs throughout the genome. `VolcanoFinder` scans were run on a sample of $n = 40$ chromosomes sampled from the present-day European population with test sites placed every 250 bp.

We find that the combination of population demography (out of Africa bottleneck) and population structure with gene flow from the African population into the Europeans leads to a reduction in power of all tests. In a non-admixed background, (Fig 6, right panel), `Volcano-Finder` (blue lines) still has moderate power to detect the adaptive introgression for $D/\theta = 3$ or 6. When divergence is low $D/\theta = 2$, as is the case for Neanderthals, `VolcanoFinder` has only low power. As in the panmictic case, admixture in the genomic background reduces the power for all four divergence values (left panel). `SweepFinder2` (green lines) obtained generally low power to detect the sweep. In contrast to `VolcanoFinder`, its power is neither affected by admixture in the genomic background nor by the divergence of the donor population. Under the most adverse conditions for `VolcanoFinder`, with low divergence of the donor ($D/\theta = 2$) and strong admixture in the genomic background, this leads to `SweepFinder2` outperforming `VolcanoFinder` in identifying (completed) introgression sweeps.

## Scans of human data

Despite the lack of contact with known archaic hominins such as Neanderthals or Denisovans, recent evidence suggests that the genomes of modern African human populations carry potentially-introgressed regions from unknown sources (*e.g.*, [60, 61]). In contrast, the genomes of non-Africans have been shown to harbor considerable levels of admixture with known archaic humans, such as Neanderthals [62, 63]. We therefore examined signals of adaptive

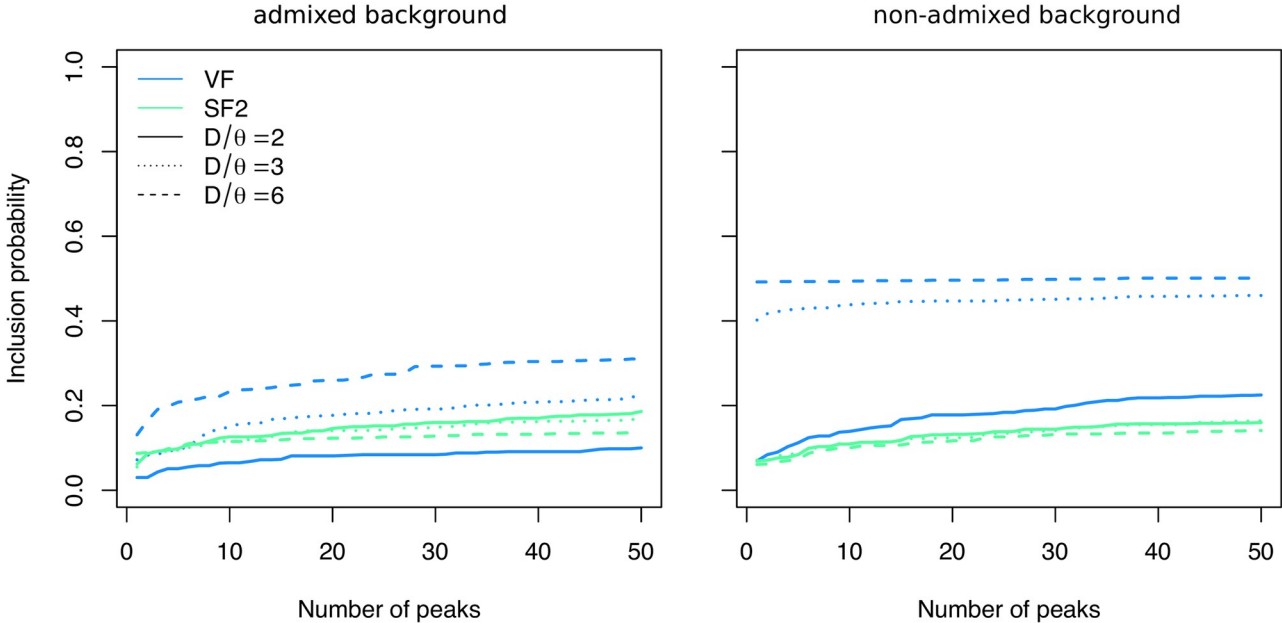

**Fig 6. Detection probability with Out-of-Africa human demography.** Probability of an introgression sweep event to be detected in a genome-scan analysis using `VolcanoFinder` (blue) or `SweepFinder2` (green). The x-axis represents the number of false-positive peaks from the neutral data that score higher than the true-positive signal. The donor species diverged from the recipient species 615 kya (solid lines) 1.230 mya (dotted), or 2.460 mya (dashed), corresponding to $D/\theta$ = 2, 3, 6, respectively. Selection on the adaptive allele is strong ($2Ns$ = 1000). The left panel shows results with an admixed genomic background while the right panel shows results with the non-admixed background. Sample size $n$ = 40 chromosomes and test sites were placed each 250 bp.

introgression in African and non-African human populations by applying `VolcanoFinder` to the Yoruban (YRI) sub-Saharan African and a central European (CEU) human populations.

In particular, we employed bi-allelic single nucleotide variant calls from the human 1000 Genomes Project [64] and polarized alleles based on alignment with the chimpanzee reference sequence [65]. To circumvent potential technical artifacts, we filtered out regions of poor mappability and alignability, and also evaluated sequencing quality at outstanding candidate regions. Furthermore, we overlaid `VolcanoFinder` scan results with an independent scan using the $T_2$ statistic of `BALLET` [48] to investigate any co-localization with evidence for long-term balancing selection. We also examined the level of nucleotide diversity ($\hat{\theta}_\pi$) across the candidate regions, as well as the level of sequence uniqueness as a more stringent measure of mappability. In the scan on Europeans, we evaluated evidence for archaic introgression at candidate regions by examining non-synonymous differences with Neanderthals [66] as well as inferred Neanderthal or Denisovan introgression segments [20, 22]. See Materials and methods for further details.

The top-scoring regions are reported in Table D1 (CEU population) and Table D2 (YRI population) of S4 Appendix. Manhattan plots of the whole genome are shown in Fig. D1 (CEU) and Fig. D2 (YRI) of S4 Appendix. In the CEU, we uncovered footprints of adaptive introgression on regions with putative Neanderthal ancestry, most notably the gene *TSHR* (Fig 7) which encodes the receptor for thyroid stimulating hormone (TSH). Using Eqs (4) and (5) with a recombination rate of $r = 10^{-8}$ recombination event per nucleotide per generation [67] and $N_e = 10^4$ [68], the inferred introgression parameters $\hat{\alpha}$ and $\hat{D}$ for the *TSHR* candidate region (Table D1 of S4 Appendix) suggest a 41.7 kb volcano centered on a 2.4 kb valley. The ratio of polymorphic sites to fixed differences in the shoulders of this volcano (175: 372) is

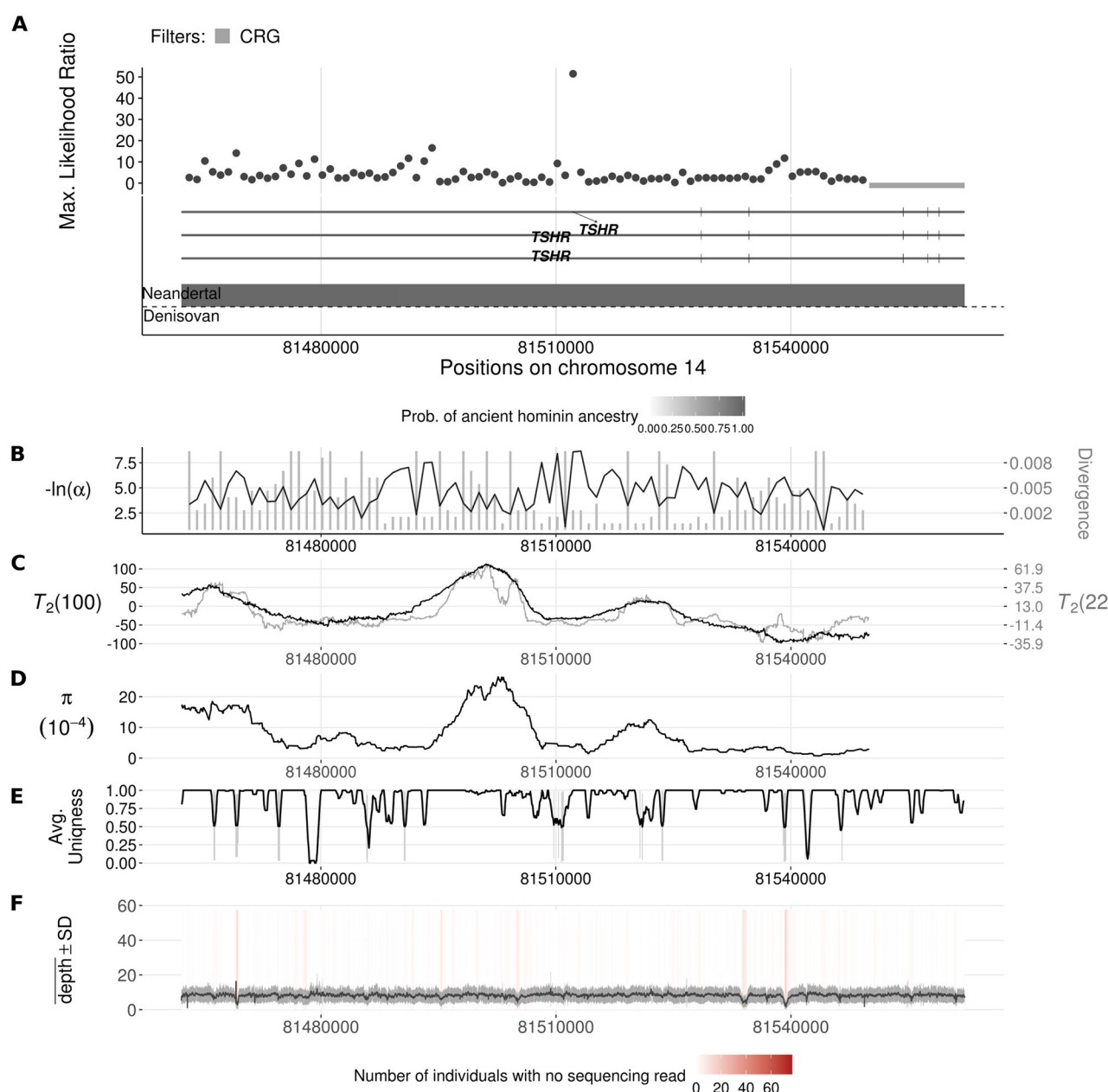

**Fig 7. Introgression sweep signals, tracks of Neanderthal or Denisovan ancestry, parameter estimates, and sequencing properties across the 100 kb region on chromosome 14 covering the *TSHR* gene in CEU. A**. Likelihood ratio test statistic computed from Model 1 of `VolcanoFinder` on data on within-CEU polymorphism and substitutions with respect to chimpanzee. Horizontal light gray bars correspond to regions that were filtered based on mean CRG. Gene tracts and labels for key genes are depicted below the plot, with the wider bars representing exons. Tracks of putative regions with Neanderthal (above the horizontal line) or Denisovan (below the horizontal line) ancestry are located below gene diagrams. Higher probabilities of Neanderthal or Denisovan ancestry are depicted with darker colored bands (data from [22]). Non-synonymous mutations with Neanderthal are indicated in red. **B**. Values for $\alpha$ and divergence $D$ corresponding to the maximum likelihood estimate of the data. Black line corresponds to $-\ln(\alpha)$ and vertical gray bars correspond to estimated $D$. **C**. Likelihood ratio test statistic computed from $T_2$ of `BALLET` on data on within-CEU polymorphism and substitutions with respect to chimpanzee using windows of 100 (black) or 22 (gray) informative sites on either side of the test site. **D**. Mean pairwise sequence difference ($\hat{\theta}_\pi$) computed in five kb windows centered on each polymorphic site. **E**. Mappability uniqueness scores for 35 nucleotide sequences across the region. **F**. Mean sequencing depth across the 99 CEU individuals as a function of genomic position, with the gray ribbon indicating standard deviation. The background heatmap displays the number of individuals devoid of sequencing reads as a function of genomic position, with darker shades of red indicating a greater number of individuals with no sequencing reads.

significantly higher than that of the genomic background (one-tailed binomial test, $p = 0.0137$) as well as that of the central region (5: 47) leading to a significant Hudson-Kreitman-Aguadé (HKA) test [69] ($p = 2.6 \times 10^{-4}$). Since divergence between Neanderthals, Denisovans and modern humans is relatively recent (4.23–5.89% of the human-chimpanzee sequence divergence [63], leading to $D \approx 1.4\theta$–$2\theta$ according to our observations) and introgressed haplotypes typically do not reach high frequency in samples of modern human populations such as CEU, we do not expect VolcanoFinder to detect most of these signals.

On the other hand, we also found outstanding candidate regions devoid of known Neanderthal or Denisovan ancestries in the scan on Europeans. One such candidate is the *CHRNB3-CHRNA6* gene cluster (Fig 8), which has been associated with substance dependence especially in Europeans (see Discussion). The inferred introgression parameters for this candidate region (Table D1 of S4 Appendix) suggest a 36 kb volcano centered on a 2.1 kb valley. Once again, the ratio of polymorphic sites to fixed differences in the shoulders of the volcano (178: 259) is significantly higher than that of the genomic background (one-tailed binomial test, $p = 2.5 \times 10^{-9}$) as well as that of the central region (5: 21) leading to a significant HKA test [69] ($p = 0.021$).

The most prominent signal across the genome in Europeans is also devoid of known archaic hominin ancestry. This region features the *APOL3* and *APOL4* (Fig 9A) on chromosome 22, which encode apolipoprotein L family proteins. The inferred introgression parameters for this candidate region (Table D1 of S4 Appendix) suggest a 20 kb volcano centered on a 0.6 kb valley. Although this region is the most prominent candidate in our analysis, the polymorphic sites to fixed differences ratio is significantly higher than that of the genomic background in the right shoulder of the volcano only (80: 145, one-tailed binomial test, $p = 0.006$). This indicates that the model-based method of VolcanoFinder relying on the whole SFS is more sensitive than the mere polymorphism:divergence ratio. The apolipoprotein L family proteins are high density lipoproteins and take part in lipid transportation [70]. They are unique to the primate lineage, and have been hypothesized to be under positive selection in humans [71]. Intriguingly, we also estimated high likelihood ratio scores around this region in the African population scan (Fig. D3 of S4 Appendix), although the peak locations in the two scans vary. Note that this candidate was not included in our final list of candidates for the YRI population (Table D2 of S4 Appendix) due to the lack of data close to *APOL4* (Fig. D3 of S4 Appendix). The concern is that test scores can be inflated near regions devoid of data. Although the breadth of the sweep as predicted by VolcanoFinder includes one such region, there is also high-quality data informing the test statistic at these sites. Furthermore, the lack of data in this region does not result in high CLR scores in CEU (Fig. D4 of S4 Appendix), lending support to the validity of the signals we observe in the scan on YRI. Instead of spanning across *APOL4* and *APOL3* like in CEU, the peak in YRI locates closer to *APOL2*, which closely neighbors *APOL1*.

In the African population scan, another interesting top-scoring region lies between the *TCHH* and *RPTN* genes on the epidermal differentiation complex (EDC) on chromosome 1 (Fig 10). This gene complex features many genes essential for the late-stage differentiation of epidermal cells and is therefore important for the integrity and functionality of skin and skin appendages [72] such as hair and nails [73, 74]. The inferred introgression parameters for the *TCHH-RPTN* candidate region (Table D2 of S4 Appendix) suggest a 23.3 kb volcano centered on a 1.3 kb valley. Although this region has the second-highest CLR in our candidate list, the ratio of polymorphic sites to fixed differences shows that the inferred shoulders are not enriched in polymorphic sites, although the HKA test [69] between the shoulders and the valley is marginally significant (64: 159 *vs*. 0: 9, $p = 0.0515$). In this case, VolcanoFinder may be sensitive to the skew in the SFS caused by the introgression sweep.

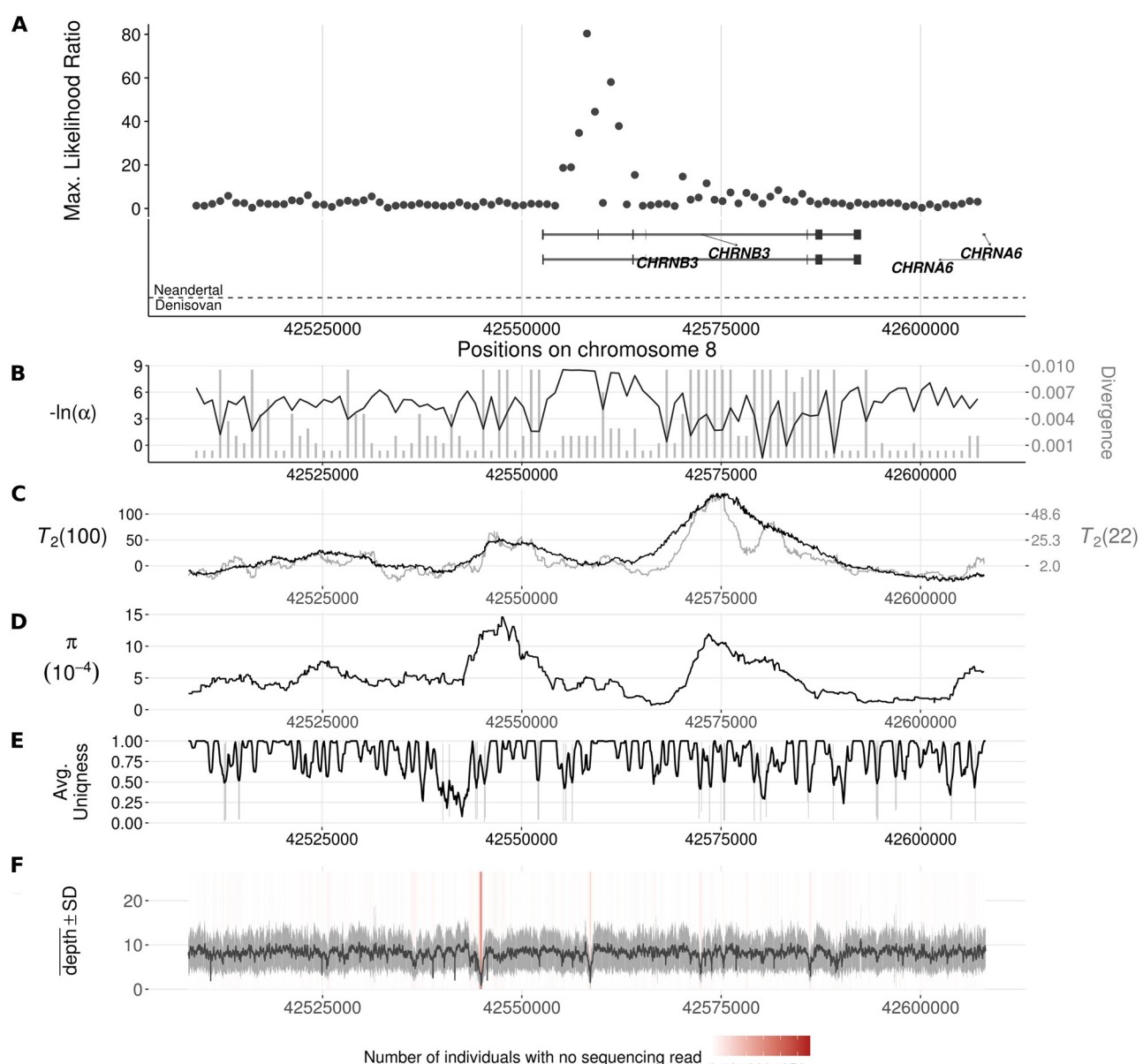

**Fig 8. Introgression sweep signals, tracks of Neanderthal or Denisovan ancestry, parameter estimates, and sequencing properties across the 100 kb region on chromosome 8 covering the *CHRNB3* gene in CEU. A**. Likelihood ratio test statistic computed from Model 1 of `VolcanoFinder` on data on within-CEU polymorphism and substitutions with respect to chimpanzee. Horizontal light gray bars correspond to regions that were filtered based on mean CRG. Gene tracts and labels for key genes are depicted below the plot, with the wider bars representing exons. Tracks of putative regions with Neanderthal (above the horizontal line) or Denisovan (below the horizontal line) ancestry are located below gene diagrams. Higher probabilities of Neanderthal or Denisovan ancestry are depicted with darker colored bands (data from [22]). Non-synonymous mutations with Neanderthal are indicated in red. **B**. Values for $\alpha$ and divergence $D$ corresponding to the maximum likelihood estimate of the data. Black line corresponds to $-\ln(\alpha)$ and vertical gray bars correspond to estimated $D$. **C**. Likelihood ratio test statistic computed from $T_2$ of `BALLET` on data on within-CEU polymorphism and substitutions with respect to chimpanzee using windows of 100 (black) or 22 (gray) informative sites on either side of the test site. **D**. Mean pairwise sequence difference ($\hat{\theta}_\pi$) computed in five kb windows centered on each polymorphic site. **E**. Mappability uniqueness scores for 35 nucleotide sequences across the region. **F**. Mean sequencing depth across the 99 CEU individuals as a function of genomic position, with the gray ribbon indicating standard deviation. The background heatmap displays the number of individuals devoid of sequencing reads as a function of genomic position, with darker shades of red indicating a greater number of individuals with no sequencing reads.

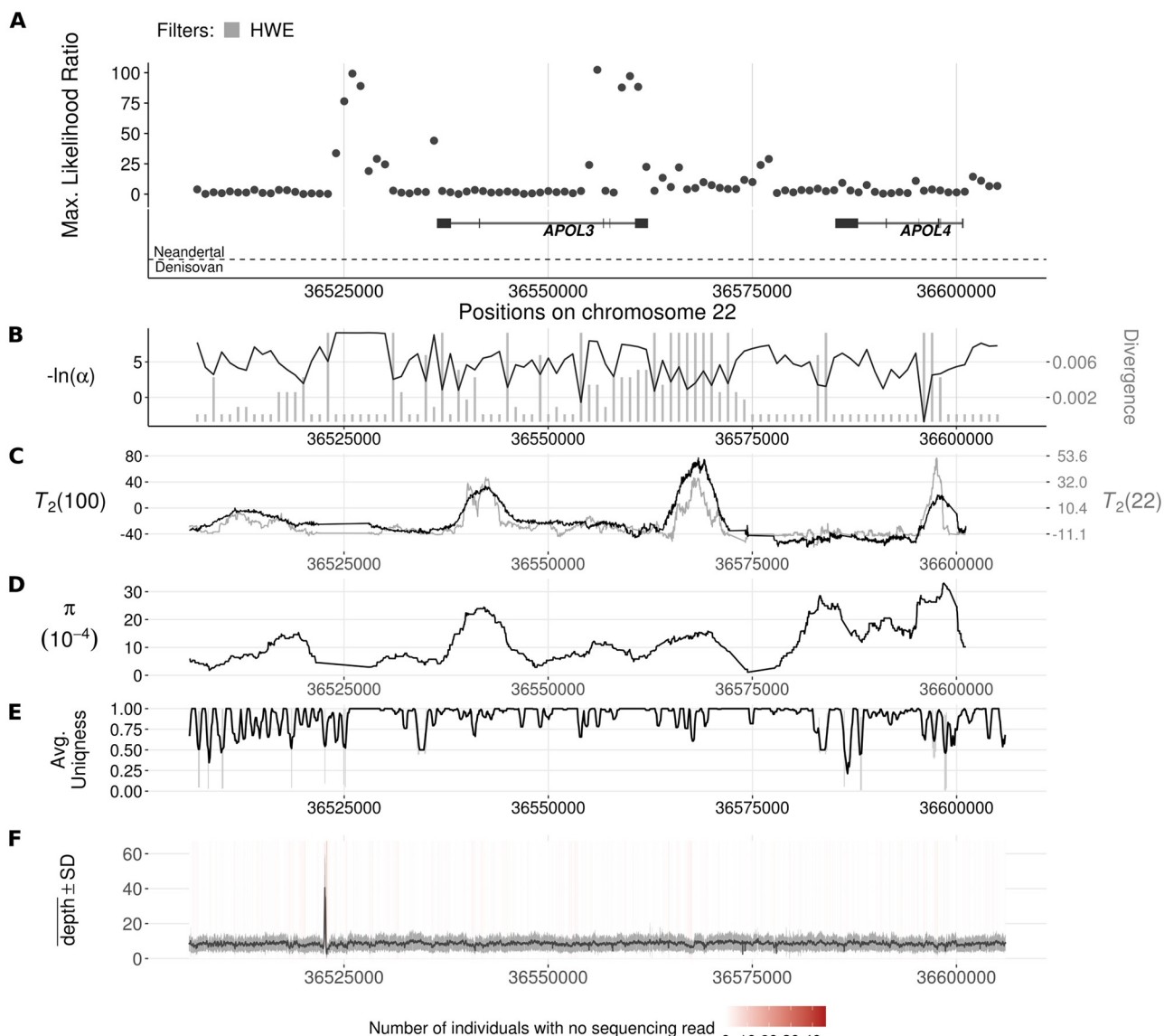

**Fig 9. Introgression sweep signals, tracks of Neanderthal or Denisovan ancestry, parameter estimates, and sequencing properties across the 100 kb region on chromosome 22 covering *APOL* gene cluster in CEU. A**. Likelihood ratio test statistic computed from Model 1 of `VolcanoFinder` on data on within-CEU polymorphism and substitutions with respect to chimpanzee. Horizontal light gray bars correspond to regions that were filtered based on mean CRG. Gene tracts and labels for key genes are depicted below the plot, with the wider bars representing exons. Tracks of putative regions with Neanderthal (above the horizontal line) or Denisovan (below the horizontal line) ancestry are located below gene diagrams. Higher probabilities of Neanderthal or Denisovan ancestry are depicted with darker colored bands (data from [22]). Non-synonymous mutations with Neanderthal are indicated in red. **B**. Values for $\alpha$ and divergence $D$ corresponding to the maximum likelihood estimate of the data. Black line corresponds to $-\ln(\alpha)$ and vertical gray bars correspond to estimated $D$. **C**. Likelihood ratio test statistic computed from $T_2$ of `BALLET` on data on within-CEU polymorphism and substitutions with respect to chimpanzee using windows of 100 (black) or 22 (gray) informative sites on either side of the test site. **D**. Mean pairwise sequence difference ($\hat{\theta}_\pi$) computed in five kb windows centered on each polymorphic site. **E**. Mappability uniqueness scores for 35 nucleotide sequences across the region. **F**. Mean sequencing depth across the 99 CEU individuals as a function of genomic position, with the gray ribbon indicating standard deviation. The background heatmap displays the number of individuals devoid of sequencing reads as a function of genomic position, with darker shades of red indicating a greater number of individuals with no sequencing reads.

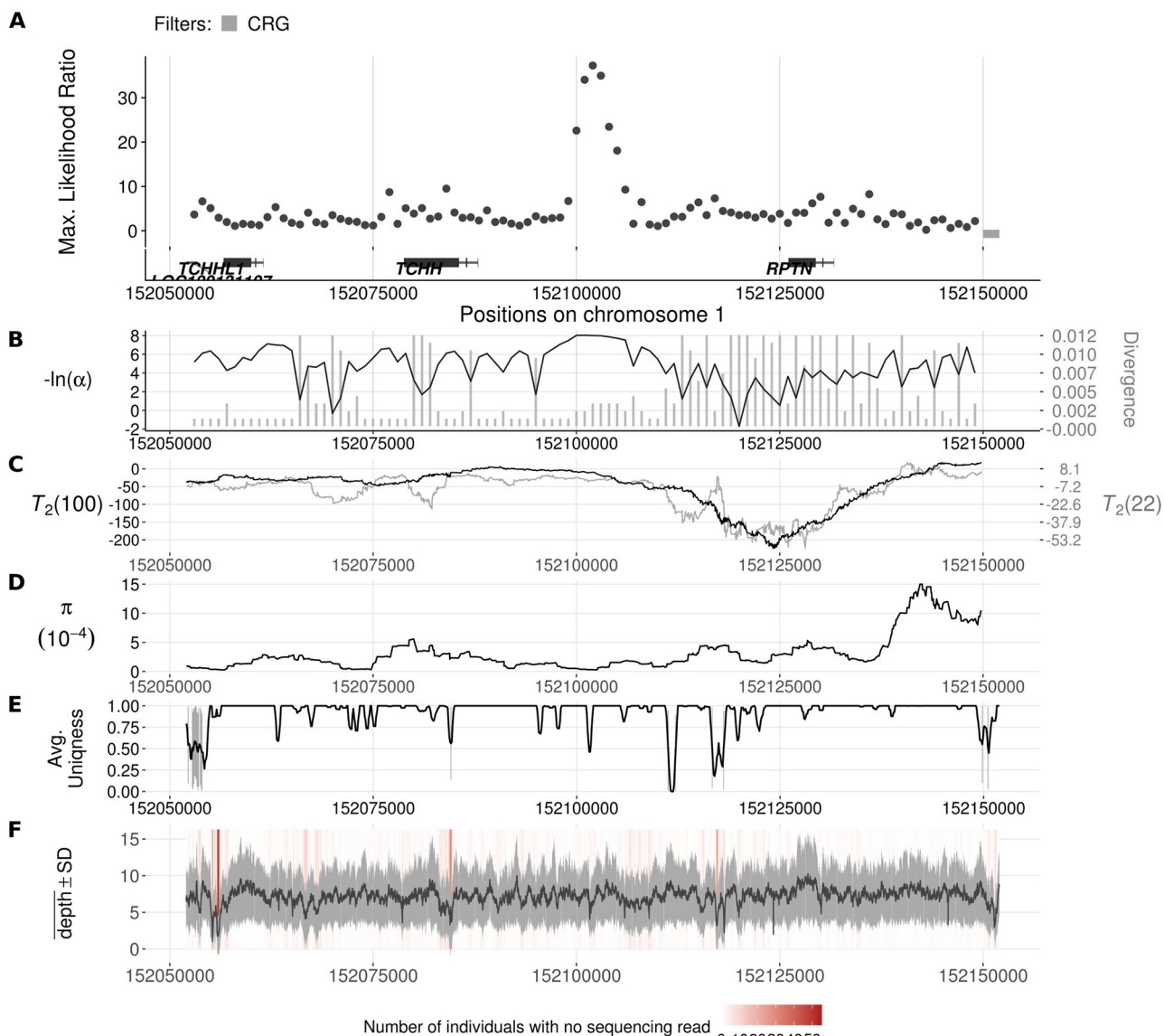

**Fig 10. Introgression sweep signals, parameter estimates, and sequencing properties across the 100 kb region on chromosome 1 covering *TCHH* and *RPTN* genes in YRI. A**. Likelihood ratio test statistic computed from Model 1 of `VolcanoFinder` on data on within-YRI polymorphism and substitutions with respect to chimpanzee. Horizontal dark gray bars correspond to regions that were filtered based on mean CRG score. Gene tracts and labels for key genes are depicted below the plot, with the wider bars representing exons. **B**. Values for $\alpha$ and divergence $D$ corresponding to the maximum likelihood estimate of the data. Black line corresponds to $-\ln(\alpha)$ and vertical gray bars correspond to estimated $D$. **C**. Likelihood ratio test statistic computed from $T_2$ of `BALLET` on data on within-YRI polymorphism and substitutions with respect to chimpanzee using windows of 100 (black) or 22 (gray) informative sites on either side of the test site. **D**. Mean pairwise sequence difference ($\hat{\theta}_\pi$) computed in five kb windows centered on each polymorphic site. **E**. Mappability uniqueness scores for 35 nucleotide sequences across the region. **F**. Mean sequencing depth across the 108 YRI individuals as a function of genomic position, with the gray ribbon indicating standard deviation. The background heatmap displays the number of individuals devoid of sequencing reads as a function of genomic position, with darker shades of red indicating a greater number of individuals with no sequencing reads.

Lastly, we also applied `VolcanoFinder` on a dataset of 500 individuals drawn uniformly at random from the global set of samples from non-admixed populations in the 1000 Genome Project dataset. However we did not find strong support for any genomic region to have undergone adaptive introgression. This result agrees with our observations that the candidate regions in the scans on African and European populations barely overlap.

## Discussion and conclusions

The hitchhiking of foreign genetic variation during adaptive introgression from a diverged donor population generates a unique volcano-shaped signature in the genetic diversity of the recipient population. Such patterns have first been described for an island model in the limit of low migration rates [38]. Here, we characterize the pattern for a scenario of secondary contact and use it to construct a genome scan method to detect recent events of adaptive introgression from sequence variation in the recipient species, without the need to know the donor species.

In sharp contrast to a classical sweep, introgression sweeps have only a narrow (expected) valley of reduced diversity around the selected site, but broad flanking regions with an excess of intermediate-frequency polymorphism relative to fixed differences to an outgroup. This excess variation is the most prominent feature of the footprint and is observed for both hard and soft introgression sweeps (*i.e.* sweeps originating from one or several hybrids, see Fig. B4 and Fig. B5 of S2 Appendix). It remains visible for extended periods of time after completion of the sweep (up to $\sim 2N$ generations, where $N$ is the effective population size).

The construction of a mathematical model for the purpose of a parametric test requires a compromise between precision and tractability. Even for the simple measure of pairwise genetic diversity, accurate predictions require approximations with several parameters to account for the variance in coalescence time during the sweep [30, 51], see also our models in the electronic supplement (S1 Appendix, Section 1). However, our results show that an extended star-like approximation with only two parameters, $\alpha$ for the strength of the sweep and $D$ for the divergence of the recipient population from the donor, offers a flexible scheme to match simulated volcano footprints for both hard and soft introgression sweeps.

The use of $\alpha$ and $D$ as flexible fit parameters poses a challenge when interpreting them as estimators for the true strength of the sweep and true divergence of the donor population. In particular, comparison with accurate approximations and simulations shows that the star-like model overestimates the predicted genetic diversity. Hence, the optimal $D$ found by `VolcanoFinder` is biased to underestimate the true divergence of the donor population.

There are further limits to the simple star-like model. Simulations show that volcano patterns are often strongly asymmetric and/or truncated due to early recombination events (Fig 3). The model also assumes that the population is sampled directly after completion of the sweep in the recipient population. Older sweep footprints may still show pronounced regions of excess variation, but could have recovered close to normal polymorphism level in the central sweep valley. More complex patterns are also expected if introgression haplotypes harbor more than a single selected allele in tight linkage. In particular, the beneficial allele can be linked to barrier genes that reduce the introgression probability and bias the footprints of successful introgression sweeps [75]. Inclusion of any such details into a statistical test would, however, require additional model parameters. For whole-genome scans, the higher-dimensional optimization that is required in this case can easily prove computationally prohibitive.

### Power analysis

The footprint of adaptive introgression combines elements of a classic selective sweep (a sweep valley) with signals that are more typical of balancing selection (excess variation at intermediate frequencies). Accordingly, we tested the power of our new method `VolcanoFinder` to detect introgression sweeps relative to two standard methods that were designed to detect classic selective sweeps (`SweepFinder 2` [47]) and long-term balancing selection (`BALLET`, [50]), respectively. In addition to ROC curves (Fig. B6 to Fig.B8 of S2 Appendix) that are typically presented in power analyses [45, 48, 49], we provide an alternative analysis that is closer

to the use of a test in a real genome scan. To this end, we estimated the probability that an introgression locus ranks among the top 1 to 50 highest CLR peaks (Fig. B3 of S2 Appendix and Fig 5) among peaks obtained from $8 \times 10^6$ CLR values from 10000 neutral replicates, which represent a whole-genome background. This approach is particularly useful for composite-likelihood tests (all three tests considered here), where standard methods for multiple-testing correction [76] that rely on independent *p*-values do not apply.

Our model postulates that an introgression sweep occurred as a result of a rare hybridization event caused by a secondary contact between diverged species (see Fig 1). In nature, admixture may often occur at a much higher rate and also affect the genomic background. We therefore explored two extreme cases: (i) a non-admixed genomic background and (ii) a neutrally admixed genomic background resulting from the same amount of admixture that allowed the introgression sweep to occur with a high probability. In natural populations, postzygotic genetic barriers [77] will typically purge part of the introgressed variation, thus reducing the genome-wide admixture to some intermediate level between these limiting cases.

In an ideal scenario, for a panmictic population with non-admixed background, `VolcanoFinder` has extremely high genome-wide power to detect introgression sweeps (test for local introgression, Fig. B3 of S2 Appendix). It clearly outcompetes the methods that have been developed for other purposes. This power is considerably reduced if the genomic background harbors higher levels of neutral admixture, in particular for weak selection. Similarly, strong population structure can lead to a further reduction in power (Fig 6). However, the detection probability remains reasonably high for strong selection and if adaptation occurs from a strongly diverged donor population ($2Ns = 1000$ in Fig 5).

Although our sweep model assumes that adaptation in the recipient population starts from a single hybrid individual, `VolcanoFinder` has virtually the same power to detect hard and soft introgression sweeps. This is in sharp contrast to the detection of classic sweeps in a single panmictic population by methods like `SweepFinder` 2. The small reduction in power for soft introgression sweeps is expected because the typical volcano patterns do not differ much between hard and soft sweeps, as explained above. We expect the same qualitative pattern also in the case of incomplete introgression sweeps, as long as the adaptive allele reaches sufficiently high frequencies > 50% in the recipient population. This suggests that `VolcanoFinder` may also detect these events with high power, but we did not test this case and quantitative predictions remain to be established.

A significant finding is the relatively high power of `VolcanoFinder` to detect old introgression sweeps. We tested this power for $T_s \leqslant 0.5$, or $2N$ generations, clearly beyond the detection limit of genome scanners for classic sweeps [49]. As an example, consider an introgression event with $2Ns = 1000$, $T_d = 4$ ($D = 9\theta$), and admixed background in Fig 5. The average probability that the introgression locus ranks among the top 50 peaks is around 66% for recent events $0 \leqslant T_s \leqslant 0.1$, but still around 33% for old events $0.1 \leqslant T_s \leqslant 0.5$. Assuming a constant rate of introgression, we expect two times as many old events than recent events because of the four times larger time window for old events. This expected enrichment in old events is even stronger with a non-admixed genomic background (Fig. B3 of S2 Appendix).

The volcano pattern is produced by the conversion of divergence into polymorphism due to the introgression event. A strong pattern (and high power of the test) therefore requires sufficient divergence between the donor and recipient populations, with $D/\theta > 3$ for our method to be powerful. Introgression sweeps from a very recently diverged donor also violates our assumption of complete lineage sorting between donor and recipient. The resulting patterns then more resemble a classical sweep and are more readily detectable with a classic genome scanner (Figs 5 and 6, and Fig. C6 of S3 Appendix). In the context of human data, we have $D/\theta < 3$ for divergence to Neanderthals or Denisovans, such that introgression sweeps may

often only leave weak volcano patterns (see also our discussion below). This may be different for introgression from so-called superarchaic hominins [78] with $D/\theta > 3$.

Several methods have been proposed to detect gene flow that could be used to identify introgressed regions (see [79] for a review). Some rely on the detection of outlier values for indicators of divergence such as $F_{ST}$ [80], Patterson's $D$ (also known as ABBA BABA, [62, 81]) or $G_{\min}$ [82]. Others are likelihood and model-based, relying on the site frequency spectrum [83, $\partial a \partial i$], hidden Markov models for the coalescent tree [84, `TreeMix`] or use approximate likelihood methods such as ABC [85]. In addition, machine-learning algorithms provide a likelihood-free approach to detecting footprints of introgression when trained using data simulated under a particular demographic model [20, 22, 25]. These methods are however not aimed at detecting the specific signature of genetic hitchhiking with an introgressed selected allele.

Like other SFS-based methods, `VolcanoFinder` assumes independence between neighbouring SNPs and is blind to strong LD patterns resulting from gene flow [86]. The rate of exponential decrease of linkage disequilibrium can be used to date admixture events [87], and sophisticated haplotype-based methods have been used to characterize admixture and selection in ancestral human populations [88, 89]. Positive selection also increases LD [90–92], and methods were proposed to employ haplotype structure to date the MRCA of a beneficial allele [93]. Haplotype-based methods are usually powerful at detecting even soft and partial classic selective sweeps [45]. In an introgression sweep, positive selection and gene flow synergistically create a pattern of long and very diverged haplotypes. Including haplotype information into `VolcanoFinder` would thus almost certainly improve its power for recent introgression events ($< 0.1N$ generations), as haplotype structure is expected to be informative over shorter time scales than patterns in the site-frequency spectrum [94]. This holds, in particular for incomplete introgression sweeps. A model extension can follow recent work, which has demonstrated that including linkage information into the CLR framework can result in a powerful test to detect incomplete classic sweeps [95]. Vice-versa, we do not expect haplotype data to contribute to an informative signal over long time scales of the order of $N$ generations. Here, further gain in testing power may be possible by changes in the underlying introgression sweep model, which currently assumes a recently completed sweep.

## Assessing evidence for adaptive introgression at empirical candidates

We applied `VolcanoFinder` to variant calls to probe for footprints of adaptive introgression in contemporary sub-Saharan-African and European human populations. With careful filters and quality-checks both before and after scans, we identified several candidate regions that may lend insights to early human evolutionary history. For application of `VolcanoFinder`, we warrant caution during data preparation and scrutiny over result interpretation, and believe it is especially important to consider factors such as the sequencing and mapping quality as well as values of other key statistics.

When preparing input for `VolcanoFinder`, we considered only regions with high mapping quality, as erroneous mapping may produce mis-matched variant calls that artificially alter the diversity of a genomic region. Specifically, following [49], we filtered 100 kb genomic segments with mean CRG100 scores less than 0.9. Such extended segments were chosen due to sweeps often affecting large genomic regions. Because `VolcanoFinder` places test sites evenly across a chromosome, for test locations within large masked regions (or in the middle of centromeres) devoid of data, the diversity levels at the edges of these regions may appear higher than expected under neutrality, coincidentally mirroring the "adaptive ridges" of increased diversity expected near an adaptive introgression allele. Consequently, test sites

within masked regions may exhibit abnormally high likelihood ratio scores. Therefore, extended genomic regions of non-missing data are desired to circumvent this potential artifact. Furthermore, due to this characteristic, output test sites should also be filtered, and it is preferable that the mask applied on the output data be more stringent than that on the input data, such that abnormalities around the filtered regions can be removed. In particular, we computed the mean CRG100 filter in 100 kb windows that overlap by 50kb. Overlapping regions in which only one of the windows passed the CRG filter were retained in the input but were excluded in the output. Moreover, we removed regions flanking telomeres and centromeres which can be difficult to map [96, 97] and may harbor increased diversity due to their repetitive nature.

After removing regions based on CRG mappability and proximity to telomeric and centromeric regions *post hoc*, we still observed that many genomic regions which passed the filters (*e.g.*, the *PTPRN2* gene region in the scan on YRI; Fig. D5 of S4 Appendix) exhibit extremely low $D$ values and high $-\log_{10}(\alpha)$. Such parameter combinations are unlikely to result from true footprints of adaptive introgression and should not be considered as genuine signals. Moreover, we noticed that these test sites often appear within or near regions devoid of data. Because gaps in input data may also be introduced in regions without mappable outgroup sequences (*e.g.*, the *PCAT*/*CEACAM4* and *B4GALNT2* regions in the scan on YRI (Fig. D6 and Fig. D7 of S4 Appendix)), we further removed test sites falling in regions with outstandingly large between-informative-site distances compared with the empirical distribution of all distances. We advise users to adopt similar screening procedures on the output data from the scan in order to exclude artifacts.

To curate the candidate regions that passed all filters, we further consulted the 35-mer sequence uniqueness scores (a more stringent measure of mappability) and the sample-wise mean sequencing read depths in order to gauge how confident we can be in the accuracy of the input data. Specifically, regions with low uniqueness may be mapped to sequencing reads from other paralogous regions and exhibit artificially high levels of variation. Sequences with low read depth may harbor unreliable variant calls, whereas those with abnormally high depth may suggest either structural variation or that sequencing reads from other regions in the genome were erroneously mapped to the region. In this light, we flagged the candidate regions with low uniqueness or abnormal mean read depths, especially when these features manifest on the lips of the "volcano" where the sequence diversity $\hat{\theta}_{\pi}$ is high. Examples of such regions are the *MUC4* (Fig. D8 of S4 Appendix) gene in the scan on CEU as well as the *CYP2B6-CYP2B7* gene region in the scan of YRI (Fig. D9 of S4 Appendix)—both regions harbor areas of low sequencing depth close to the CLR peak. The *MUC4* candidate region was discarded from our final list because of the neighbouring CRG-filtered region (Fig. D8 of S4 Appendix) whereas the *CYP2B6-CYP2B7* gene region passed this filtering step and is actually the top candidate in the YRI list (Table D2 of S4 Appendix). We advise users to consider such candidates with caution and possibly look for other evidence of an introgression sweep. As far as the *CYP2B6-CYP2B7* gene region is concerned, the polymorphism and divergence pattern in the CEU sample shows no support for an introgression sweep either, suggesting that this introgression signal might be an artifact.

Although the sequencing read depth and sequence uniqueness alone are insufficient to determine whether the observed high likelihood ratios are the result of artifacts, reasonable read depths and high sequence uniqueness nonetheless provide strong support that the footprints observed at candidate regions are genuine. To provide additional support for footprints of adaptive introgression, we also consulted the values of the BALLET $T_2$ statistic at putative adaptive regions. Because $T_2$ is sensitive to ancient balanced alleles, it may report slightly

elevated scores for introgressed regions and low scores for sweep regions. Therefore, in putative adaptively-introgressed regions, we should not only see high likelihood ratios reported by VolcanoFinder, but also expect to see a concomitant dip in $T_2$ scores, consistent with the "volcano"-shaped footprint of nucleotide diversity. We are able to find these supporting features in *TSHR*, *CHRNB3*, and *APOL3* gene regions in the scan on CEU (Figs 7, 8 and 9, respectively), as well as the *TCHH-RPTN* intergenic region in YRI (Fig 10).

Lastly, we noticed that there are previously-identified candidate regions for adaptive introgression that do not overlap with our lists of top candidates. In particular, we did not recover *BNC2* [98], the *OAS1/2/3* gene cluster [99–102], and the *TLR1/6/10* gene cluster [101–103] in Europeans, as well as *KCNIP4* and *TRPS1* recently reported in sub-Saharan Africans [104]. Upon closer inspection, we found that the majority of these regions were removed by our mappability filter prior to the application of VolcanoFinder, which at least partly explains why they escaped detection. Moreover, the discoveries of these candidate genes were mainly based on known sequences of archaic humans and were guided by introgression-signaling statistics based on allele- or haplotype-sharing [105–108], by modeling the demography of introgression agnostic of selection [98, 109–112], or by machine learning approaches [104, 113–116]. These methods do not jointly model the effect of introgression and sweeps on genetic variation, and often determine the adaptive candidates based on allele- or frequency cutoffs after the search for genomic segments of archaic ancestry. These features make them typically adept at discovering recent introgression events, in which the sweep often would have had insufficient time to complete. Although the machine learning approaches do jointly consider multiple components of genetic variation, and some [104, 116] do not rely on the sequences from donor populations, the demographic models followed by their simulated training datasets are explicitly based on inferences of known archaic humans, and therefore are not truly agnostic of the donors.

The approach employed by VolcanoFinder substantially differs from these past studies not only in that it does not rely on the data from a putative donor population, but also that it explicitly uses an evolutionary model to generate an expectation of genetic diversity during an adaptive introgression event. Further, due to modeling assumptions, VolcanoFinder prefers certain introgression scenarios over others, and the previous candidates, albeit some being well-characterized (e.g., *BNC2*), may not have prominent key features that VolcanoFinder uses to discriminate between adaptive introgression events and neutrality. In particular, VolcanoFinder is more sensitive to strong complete sweeps (forming the "valley" of a volcano) from a highly diverged donor (the "edges" of a volcano), especially in the presence of background introgression. Hence, even if the previously-reported candidate regions survived our filters, it is still sensible that they may not lead to a high VolcanoFinder likelihood ratio score, as the divergence between the donor and recipient may not be distant enough or the selection strength may not be strong enough. In fact, among all the previously-identified candidates, the remaining region of *BNC2* (Fig. D10 of S4 Appendix) scored the highest in CEU, at 18.272, lower than our cutoff value of 20 for candidates in the CEU and YRI scans. Though the optimal divergence value is approximately 0.003034, which is similar to other candidates, we only observe a minor "volcano" pattern from the $\pi$ values flanking the region (Fig. D10 Panel D), implying that the spatial footprint is much less conspicuous than other candidates we reported. These results suggest that the putative introgression sweep at *BNC2* may not be as intense or complete as other candidates, which is supported by the archaic haplotype (inferred as Neanderthal) being at a frequency of roughly 0.7 in Europeans [107, 113], suggesting that a putative adaptive introgression sweep in this region was incomplete and stemmed from a recently-diverged donor population.

## Implications of the `VolcanoFinder` scans in Europeans and Africans

After careful screening and curating of the candidate genes from our scans on contemporary Europeans (CEU) and sub-Saharan Africans (YRI), we reported 27 candidate regions in CEU and 7 candidate regions in YRI. With out-of-African populations having more contact with archaic hominins, it is sensible that we are identifying a greater number of candidate genes in Europeans than in Africans. Among the candidate regions reported, we found the *TSHR*, *CHRNB3*, and *APOL4* gene regions particularly interesting in CEU, and the *TCHH-RPTN* region highly interesting in YRI. Meanwhile, the lack of strong support for any genomic region in the scan on the pooled global population indicates that detectable adaptive introgression events with other hominins prior to the migration out of Africa may be unlikely.

In CEU, we found both strong evidence for adaptive introgression and Neanderthal ancestry in the *TSHR* gene (Fig 7). This gene encodes the receptor for TSH, or thyrotropin, the pituitary hormone that drives the production of the thyroid hormones [117]. In addition to its pivotal role in thyroid functions and the thyroid-mediated energy metabolism in most tissues, the TSH receptor has also been shown to take part in skeletal remodeling [118, 119], epidermal functions and hair follicle biogenesis [120–122], gonad functionality [123], as well as immunity [124]. Moreover, accumulating evidence also show its expression in adipose tissues [125, 126], and that it can regulate lipolysis [127, 128] and thermogenesis [126, 129]. Considering the contrasting climates of Europe and Africa, we speculate that the selective pressure on the *TSHR* gene in Europeans may be explained by the need to update their thermo-regulation in response to the colder climate. As the Neanderthal would have been better adapted to the local environment by the time humans expanded out of Africa, it is also sensible that this genomic region carries considerable Neanderthal ancestry (Fig 7A).

In contrast, the second highest candidate, the *CHRNB3* gene region, does not carry substantial Neanderthal ancestry (Fig 8). This gene encodes a nicotinic cholinergic receptor, and modulates neuronal transmission on synapses. Multiple genetic variants on this locus have been repeatedly associated with substance dependence, including smoking behavior [130, 131], nicotine dependence [132, 133], alcohol consumption [134, 135], and cocaine dependence [135]. Furthermore, in cross-ethnicity studies, SNPs on this locus not only have higher allele frequencies in non-African populations [132], but also have a smaller effect on the nicotine dependence behavior in African Americans than European Americans [133]. The absence of Neanderthal or Denisovan ancestry around the footprints of adaptive introgression, the moderate inferred divergence value $\hat{D}$ (Fig 8B, $\hat{D} = 0.023 = 3\hat{\theta}_\pi$), as well as the higher allele frequencies in non-Africans relative to African populations, may suggest a possible encounter with an unknown archaic hominin population during the out-of-Africa migration.

Also devoid of known archaic hominin ancestry, the top candidate *APOL* gene cluster in CEU (Fig 9 and Fig. D1 of S4 Appendix) not only exhibits substantially higher CLR scores than other candidates, but also shows evidence for adaptive introgression in YRI. Our closer inspection show that the peaks in the two scans do not co-localize, with the peak in CEU spanning *APOL3*, whereas the peak in YRI locating closer to *APOL2*, which closely neighbors *APOL1*. A potential interpretation for this observation is that an introgression event predated the split of African and non-Africans, with variants around *APOL2* and *APOL1* advantageous to the local environments of Africans, whereas other variants between *APOL4* and *APOL3* on a different haplotype were subject to a different source of selective pressure in non-Africans. In line with this interpretation, in addition to its influence on blood lipid levels [136], APOL1 can also form pores on lysosomes after being engulfed and kill *Trypanosoma* parasites [137]. The *Trypanosoma* are known for causing sleeping sickness (*i.e.*, trypanosomiasis) and have been rampant in Africa [138, 139]. Moreover, though some subspecies of *T. brucei* have

evolved to be resistant to it, some genetic variants unique in African human populations have been shown to counteract their defense [140, 141]. In the absence of this pathogenic threat, however, enhancing APOL1's trypanosome lytic activity in turn elevates the risk of cardiovascular diseases and chronic kidney diseases [141–143]. These diverse features of the *APOL* gene cluster may provide a biological basis for distinct selective pressures in Africans and non-Africans.

Further echoing the recent evidence for archaic introgression in African humans, we found strong evidence for the *TCHH-RPTN* region in YRI to carry footprints of adaptive introgression. The gene *TCHH* encodes trichohyalin, a precursor protein crosslinked with keratin intermediate filaments in hair follicle root sheaths and hair medula [144–147], and is crucial for hair formation [146, 148]. In fact, SNPs in this gene have been associated with straighter hair in Europeans [146, 149], as well as Latin Americans [150]. The gene *RPTN*, on the other hand, encodes repetin, another keratin filament-associated protein expressed in skin [151]. Although its exact biological role awaits further elucidation, probably due to its relatively recent discovery, an increase of *RPTN* expression was observed in clinical cases of atopic dermatitis [152]. Further, variants in *RPTN* have also been recently reported to also associate with straight hair in both Europeans and East Asians [153]. In African populations, although it is suggested that variation in curly hair is a complex trait that involves many genes, *TCHH* is among the candidate genes [147]. The footprints of adaptive introgression on this locus therefore imply a potential setting in which the ancestors of contemporary African populations acquired the adaptive alleles from a possible admixture with an unknown archaic hominin, resulting in, at least, beneficial phenotypes of hair morphology and curvature.

Taken together, our scans for adaptive introgression on two human populations have not only recovered candidate regions in Europeans that align with previous observations of Neanderthal and Denisovan ancestry (*e.g.*, *TSHR*), but also revealed novel candidates in both Europeans and Africans that locate in regions without evidence for introgression from known archaic hominins. These results lend insights on the environmental selective pressure, such as lipid and energy metabolism and pathogen defense, that may have acted on early humans. Furthermore, together with the inferred divergence time as well as the reference of introgressed regions from known archaic hominins, we have assembled a set of clues related to the distribution of as-yet-unknown archaic humans and their interactions with our ancestors.

## Materials and methods

### Footprints of adaptive introgression: Forward simulations

We used two distinct simulation approaches. The accuracy of the analytical predictions of the model was first studied using a mixed forward and backward method (described here) that fully simulates the stochastic trajectory of the selected allele, initially introduced in a single lineage. The power analysis was conducted in a second stage using a fully coalescent-based method (described in the section on power analysis below) that does not allow for direct control of the number of introgressed lineages, but enables to easily simulate hard and soft introgression sweeps, as well as to assess the effect of the genome-wide admixture resulting from secondary contact.

Due to the long divergence time, individual-based forward-time simulations of the full model are computationally expensive and time limiting. While the current coalescent-based method msms [154] can incorporate the effects of selection at a single locus, demography cannot be included when conditioning on the fixation of the foreign adaptive allele. This is because we cannot guarantee that, backward in time, the sweep will have completed before the allele returns to the common ancestral population.

To simulate the full model efficiently, we use a backward-time, forward-time approach. The coalescent simulator msprime [54] is capable of quickly simulating large genomic regions for even whole-population-sized samples. We use this to implement the model of divergence without gene flow among the donor and recipient populations, as well as a third distant outgroup. Sampling one lineage from the outgroup to polarize the data, we sample $2N - 1$ lineages from the recipient population and one lineage from the donor population to form a diploid population containing a single hybrid individual. We then import the data to simuPOP [155]. In the foreign haplotype of the hybrid individual, we place at the center of the sweep region a beneficial allele with selective strength $s$. We repeatedly run the evolutionary model forward in time until an iteration with a successful sweep is found.

We simulate a genomic region that spans $R = rd = s$ left and right of the benefical mutation, as this covers the region where genetic diversity is increased. However, for computation speed, we do not simulate a continuous genome, but rather a set of 100-bp intervals centered at distance $R$. Here, the recombination rate per site $r$ is low so that recombination within the windows is unlikely, but recombination between the windows occurs with appreciable chance. This ensures that the mean expected heterozygosity calculated for a given window is representative of the genealogical distribution specifically at that site. Furthermore, the mutation rate per site $\mu$ is chosen so that, even with high divergence, multiple mutation hits at a single site are unlikely.

## Software implementation

VolcanoFinder is implemented in the C programming language using much of the code base in SweepFinder2 [50] as its foundation. The software takes in data on derived allele counts at biallelic sites ordered along a chromosome, employs information either on polymorphic sites or on both polymorphic sites and substitutions, and implements one of the four model combinations introduced here (Model 1 or 2, with or without fixed differences). The software also requires as input the empirical mutation frequency spectrum, which it uses as the null hypothesis in the composite likelihood calculation (as in [47]).

The user defines the number of test sites over which to compute the composite likelihood ratio test statistic, and these test sites are evenly spaced across the input genomic region or chromosome. Note that this implies that a test site does not need to be located on any particular data point. At a particular test site, VolcanoFinder searches a grid of divergence values $D$ separating the donor and recipient populations and, for each, optimizes over the sweep strength $\alpha$. By default, $D$ is optimized over the grid $D \in \{\hat{\theta}_\pi, 2\hat{\theta}_\pi, \ldots, k\hat{\theta}_\pi\}$ under Model 1 and $D \in \{2\hat{\theta}_L, 3\hat{\theta}_L, \ldots, k\hat{\theta}_L\}$ under Model 2, where $k$ is chosen as the maximum positive integer with $D \leqslant 2D_o$ if fixed differences are polarized or $D \leqslant D'_o$ if they are not. Here, $D_o$ is the divergence between the recipient species and its MRCA with the outgroup species, $D'_o$ is divergence between the recipient species and the outgroup, $\hat{\theta}_\pi$ is Tajima's estimator of the population-scaled mutation rate $\theta$, and $\hat{\theta}_L$ is another unbiased estimator of $\theta$. These values are computed internally in the software from the unnormalized site frequency spectrum, with $D_o$ (or $D'_o$) computed as $S_1(1)$ (Eq (12) with polarized or non-polarized fixed differences), $\hat{\theta}_\pi = S_1(2)$, and $\hat{\theta}_L = \frac{1}{n-1}\sum_{i=1}^{n-1} iS_i(n)$. Users are able to specify the set of $D$ values that they wish to cycle over instead.

Note that although VolcanoFinder can use all available data on an input chromosome to compute a composite likelihood ratio at a given test site, data points far from the test site will not alter this likelihood ratio, as the site frequency spectrum expected for such distant sites will be the same under neutrality as for adaptive introgression. For this reason, we follow the

implementation used in `SweepFinder` [47] and cut the computation off when data points are distant enough from the test site. That is, we restrict the computation to data points in which $\alpha d \leqslant 12$, where $d$ is the distance between the test site and a given data point. Furthermore, though the sweep strength parameter matches that of the original `SweepFinder` model [47], we found that the hard-coded limits on $\alpha$ in the `SweepFinder` implementations [47, 50] prevent the software from accurately detecting sweeps of that size. `SweepFinder` still has high power to observe the patterns of a classic hard sweep, identifying a model that underestimates the true strength of selection. Because `VolcanoFinder` relies on information further to the periphery of the sweep region, this generated a loss of power to detect sweeps. We therefore reduced the minimum $\alpha$ considered by `VolcanoFinder` by an order of magnitude compared to `SweepFinder` so that wide volcano patterns (*i.e.*, large $d$) can be observed by our method.

Because `VolcanoFinder` is computationally intensive, we provide several features in the software that allow introgression scans to run in parallel. First, for a given input dataset, the user can choose a number $m$ such that the dataset is broken into $m$ blocks of test sites with an equal number of contiguous test sites in each block. `VolcanoFinder` can then be applied to the same dataset $m$ times, where it computes the values across the sites in one of the $m$ blocks in each application. These blocks of contiguous test sites can then be scanned separately on different cores, and an auxiliary script will merge the $m$ scans into a single scan. In addition, for some users such a fine grid of $D$ values may be unnecessary. To this end, the software also implements an option for specifying a single user-defined value for $D$—allowing to easily scan for adaptive introgression with many values of $D$ simultaneously in parallel.

## Power analysis

**Model and simulation procedure.** Coalescent simulations were performed with `coala` [156] as a frontend to `msms` [154]. We assume $n = 40$ lineages are sampled from a focal species and one lineage is sampled from an outgroup that diverged at time $T_{sp} = 10$ units of $4N$ generations in the past. Detailed descriptions are given in S2 Appendix, Section 2.

For introgression sweeps, we model a secondary contact (Fig. B1 S2 Appendix) where the recipient (focal) and donor (unknown) species diverged at time $T_d < T_{sp}$ and a beneficial allele with selection coefficient $s$ was introgressed from the donor into the recipient species during a short pulse of migration. The size of the donor species is adjusted either to enforce a hard introgression sweep or to allow the introgression of neutral polymorphism from the donor species, possibly leading to soft introgression sweeps. The migration parameters (migration rate, time and duration) are adjusted such that the fixation probability of the beneficial allele in the recipient species is high ($\pi_{\mathrm{fix}} = 0.95$) and the introgression sweep ends at time $T_s$ (see details in S2 Appendix, Section 1). We assessed the effect of the divergence time ($T_d \in \{1, 2.5, 4, 5.5\}$, *i.e.* $D/\theta \in \{3, 6, 9, 12\}$), the ending time of the introgression sweep ($T_s \in \{0, 0.1, 0.25, 0.5\}$) and the selection coefficient ($2Ns \in \{100, 1000\}$) for hard and soft introgression sweeps, leading to 64 parameter sets. Neutral coalescent simulations without admixture (one parameter set) or with the same level of admixture (64 parameter sets) were used as neutral references.

Coalescent simulations under three demographic models involving balancing selection (overdominance, Fig. B2 of S2 Appendix) were also conducted to assess the robustness of `VolcanoFinder` to excess expected heterozygosity in the focal species caused by long term balancing selection starting at time $T_s$. Combining six values for $T_s \in \{1.25, 5, 8.75, 12.5, 16.25, 20\}$ and three demographic models leads to 18 parameter sets. Neutral coalescent simulations with the same demographic model (three parameter sets) were used as neutral references.

**Statistical methods for power estimation.** A detailed description is provided in S2 Appendix, Section 3. The genome-wide reference backgrounds used by all composite likelihood methods were obtained from neutral coalescent simulations.

For each simulated sequence, genome scan methods provide a list of locations for the selected locus and composite likelihood ratios. The maximum LR over a simulated sequence (or possibly in a smaller region) was used as a test statistics. For each parameter set, the null distributions of the test statistics were obtained from 10 000 neutral replicates and the rejection rates for increasing false positive rates (up to 5%) were estimated from 1 000 non neutral replicates.

In the case of introgression sweeps, two kinds of neutral references were used in separate analyses: either a non-admixed reference background (common to all parameter sets) or an admixed reference background (one per parameter set) with the same migration parameters as the associated non-neutral case. This enables to consider the two limiting cases where introgressed alleles are either quickly purged by natural selection (non-admixed background) or behave fully neutrally (admixed background).

The detection probability of an introgression sweep in a genome-wide study focussing on top candidates was estimated as the proportion of the 1 000 non-neutral replicates for which the highest LR would rank in the genome-wide top 50 peak values obtained under neutrality. Peak values were obtained from the $8 \times 10^6$ LR values generated by 10 000 neutral replicates. Neighbouring peaks (separated by less then 10 LR values) were merged.

## Human data analysis: Materials and methods to generate the CEU and YRI data

For each human population analyzed in this study, we used genotypes from variant calls of the 1000 Genomes Project Phase 3 dataset [64]. Alleles were polarized as derived or ancestral based on the allelic state in the aligned chimpanzee (panTro5) reference genome [65], and only mono- or bi-allelic single nucleotide sites that could be polarized were considered. As in [49], to ensure that we only used sites in regions of high mappability and alignability, we examined the mean CRG 100mer score for each 100 kilobase (kb) genomic region whose centers are spaced every 50 kb apart, and only considered sites in regions with a mean score no lower than 0.9.

Based on the filtered data, we summarized the non-normalized site frequency spectra for each population analyzed, and computed the per-site heterozygosity $\hat{\theta}_\pi$ across 108 Yoruban (YRI) and 99 European (CEU) individuals to be 0.001004392 and 0.0007584236, respectively, which is in line with previous estimates of the mutation rate [157]. Furthermore, from these frequency spectra, we also computed each population's divergence $D'_0$ with chimpanzees as 0.01251347 and 0.01251496 for YRI and CEU, respectively, which is also in line with prior estimates [157]. The genome-wide proportion of polymorphic sites among informative sites (polymorphic sites and fixed differences) was 0.3905585 and 0.2763345 for YRI and CEU, respectively. We applied `VolcanoFinder` on their genomic data accordingly, placing a test site every one kb across each autosome. To mask the test sites falling in regions with missing or potentially problematic data, we removed from downstream analyses test sites in the aforementioned 100 kb windows with mean scores lower than 0.9, as well as test sites within 100 kb of a centromeric or telomeric region.

Candidate loci were defined as showing a peak of CLR values. We used a minimum ln (CLR) of 20 and a minimum distance of 15 kb between peaks. In order to remove artifactual candidates, we discarded candidates that stood in regions depleted of informative sites (the minimum distance to the nearest informative site had to be lower than the 0.9995 quantile of

the distribution of the distances between consecutive informative sites on the same chromosome) and only retained candidates for which the inferred selection parameters were compatible with the typical volcano footprint of an introgression sweep ($\hat{D} > \hat{\theta}_\pi$ and a volcano half-width, inferred from $\hat{\alpha}$, $\hat{D}$ and Eq (5), larger than 5 kb). Eq (5) suggests that this minimum half-width corresponds to a compound selection parameter $2N_e s \geqslant 2.7$ given realistic values for the recombination rate and the effective population size in humans, $r = 10^{-8}$ recombination events per nucleotide per generation [67] and $N_e = 10^4$ [68]. Such a low value enables us to take into account the variance of local recombination rates and the intrinsic trend of `VolcanoFinder` to underestimate the selection coefficient for old introgression events (Fig. B17 and Fig. B18 of S2 Appendix).

To further curate empirical candidates, we generated the sequencing coverage based on the BAM files of each individual included in the dataset for a particular population (YRI or CEU). For each population, sample-wide mean sequencing depth and the corresponding standard deviation were computed and used as a reference for assessing candidate regions. As a complementary measure, we also considered the number of individuals devoid of sequencing reads at a particular genomic position to further examine data quality. Furthermore, we examined the mappability uniqueness of each 35 nucleotide sequence (data from [158]; accessed via UCSC Genome Browser) for all candidate regions. This criterion can further flag potential issues with sequence mapping. Moreover, to investigate potential sources of introgression, we also examined the non-synonymous differences between modern humans and Neanderthals [66], as well as the regions of mapped Neanderthal or Denisovan introgression segments that intersect candidate regions in the CEU population [20, 22]. Moreover, to investigate whether introgression sweep signals co-occur with signals of ancient balancing selection, we applied the $T_2$ statistic from `BALLET` [48] to the same polymorphism and substitution data on which `VolcanoFinder` was applied, and filtered the output with the same filters we applied to `VolcanoFinder` output.

For the CEU scan, to gain insight about previously-hypothesized introgression events inferred from Neanderthal and Denisovan genomes, we consulted the haplotype tracks for archaic ancestries inferred by [20, 22], of CEU and western Eurasian populations, respectively. We also obtained the Neanderthal alleles at protein-coding differences between humans and chimpanzees ([66]; downloaded from UCSC Genome Browser in September 2018). Note that because [66] considered the protein sequences between humans, chimpanzees, and orangutans when determining coding differences specific to the human lineage and also only considered non-synonymous substitutions, the dataset only contained information on non-synonymous coding differences in genes with matching numbers of homologs in the human and chimpanzee lineages.

Finally, in order to characterize a predicted increase of the polymorphism:divergence ratio in the shoulders of candidate volcanoes of introgression, the counts of polymorphic sites and fixed differences were obtained in the inferred valley and shoulder regions (distances from the LR-peak given by $\hat{\alpha}$, $\hat{D}$ and Eqs (4) and (5)). The polymorphism:divergence ratios in the volcano shoulders were compared to that of the genomic background using a one-tailed binomial test and to those of their central valley in a Hudson-Kreitman-Aguadé (HKA) test [69].

## Supporting information

**S1 Appendix. Model and analysis.** Supplementary material for the model and analysis, organized into the following sections: (1) Analytic model (2) Model two—accounting for coalescent time within the recipient population (3) The SFS after the introgression sweep (4)

Comparison of models one and two (5) Performance of VolcanoFinder and SweepFinder models.
(PDF)

**S2 Appendix. Power analysis—Large genome.** Supplementary material for the power analysis within the context of a large genomic background, organized into the following sections: (1) Adjusting the migration rate and duration during the sweep (2) Coalescent simulations (3) Genome scans (4) Statistical power in non-admixed genomic background (5) Probability of detecting an introgression sweep in an outlier study (6) Inferred parameters of the selection model.
(PDF)

**S3 Appendix. Power analysis—Large chromosome.** Supplementary material for the power analysis within the context of a large chromosomal region, organized into the following sections: (1) Peak identification and assessing power (2) Power analysis—10 Mb chromosomes (3) Robustness to classic sweeps (4) Robustness to background selection (5) Chimeric chromosomes.
(PDF)

**S4 Appendix. Human data.** Supplementary material for the human data analysis. This includes (1) The Manhattan plots for YRI and CEU (2) The table of candidate loci for YRI and CEU (3) The additional genomic regions that are highlighted in the main text.
(PDF)

# Acknowledgments

The computational results presented have been achieved in part using the Vienna Scientific Cluster (VSC). Portions of this research were conducted with Advanced CyberInfrastructure computational resources provided by the Institute for CyberScience at Pennsylvania State University.

# Author Contributions

**Conceptualization:** Rasmus Nielsen, Joachim Hermisson.

**Data curation:** Xiaoheng Cheng, Michael DeGiorgio.

**Formal analysis:** Derek Setter, Sylvain Mousset, Joachim Hermisson.

**Funding acquisition:** Michael DeGiorgio, Joachim Hermisson.

**Investigation:** Derek Setter, Sylvain Mousset, Xiaoheng Cheng.

**Methodology:** Derek Setter, Sylvain Mousset, Michael DeGiorgio, Joachim Hermisson.

**Project administration:** Sylvain Mousset, Joachim Hermisson.

**Resources:** Michael DeGiorgio.

**Software:** Derek Setter, Sylvain Mousset, Michael DeGiorgio.

**Supervision:** Michael DeGiorgio, Joachim Hermisson.

**Validation:** Derek Setter, Sylvain Mousset, Xiaoheng Cheng, Joachim Hermisson.

**Visualization:** Derek Setter, Sylvain Mousset, Xiaoheng Cheng.

**Writing – original draft:** Derek Setter, Sylvain Mousset, Xiaoheng Cheng.

**Writing – review & editing:** Derek Setter, Sylvain Mousset, Xiaoheng Cheng, Rasmus Nielsen, Michael DeGiorgio, Joachim Hermisson.

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
