## [Decision Letter · Decision Letter 0]

3 Sep 2019

Dear Dr Setter,

Thank you very much for submitting your Research Article entitled 'VolcanoFinder: genomic scans for adaptive introgression' to PLOS Genetics. Your manuscript was fully evaluated at the editorial level and by independent peer reviewers. The reviewers appreciated the attention to an important problem, but raised some substantial concerns about the current manuscript. Based on the reviews, we will not be able to accept this version of the manuscript, but we would be willing to review again a much-revised version. We cannot, of course, promise publication at that time.

If you decide to revise the manuscript for further consideration at PLOS Genetics, please aim to resubmit within the next 60 days, unless it will take extra time to address the concerns of the reviewers, in which case we would appreciate an expected resubmission date by email to plosgenetics@plos.org.

[LINK]

We are sorry that we cannot be more positive about your manuscript at this stage. Please do not hesitate to contact us if you have any concerns or questions.

Yours sincerely,

Alex Buerkle

Associate Editor

PLOS Genetics

Bret Payseur

Section Editor: Evolution

PLOS Genetics

This manuscript has been carefully reviewed by three referees, each of whom notes that this is a timely contribution to our understanding of archaic adaptive introgression. Each makes a number of substantial suggestions for improvement of the manuscript. To these I will add that some readers will appreciate greater clarity on the timescale that is being considered, as this modeling and analysis pertains to archaic introgression, as opposed to research on adaptive introgression in contemporary, early generation hybrids. Along these lines, while the contrasts among software tools are interesting, I agree with a reviewer comment that it would be worthwhile to state a bit more emphatically the limits to learning about adaptive introgression. These suggestions aside, this is an impressive study that will be strengthened by considering and incorporating suggestions from the review process.

Reviewer's Responses to Questions

**Comments to the Authors:**

Reviewer #1: The authors present an excellent manuscript describing, testing, and applying an approach to detect adaptive archaic introgression. I only have a few comments about this strong manuscript:

1. Software: Copying and pasting the software example gives me the following error on a Mac OS X 10.12.6 (High Sierra) (this was observed in all examples in the manual):

../VolcanoFinder -i 800 psvf_2293_0242.txt spectvf_2300.txt -1 0 1 vf_2293_0242_2300.out

You have chosen to get introgression sweeps using pre-computed frequency spectra

done readsnps datasize=3059 nmax=40 nmin=40 xmax=41 invar=1

Initializing binomial coefficients

findsweeps smin=5.100000e+01 smax=9.999000e+04 gridsize=800 minlike=-inf

calcprobs nmax=40 nmin=40 xmax=41 invar=1

done calcprob

Assertion failed: (pr >=0.0 && pr<1.00000001), function ln_likelihood_introgression, file VolcanoFinder.c, line 980.

Abort trap: 6

2. Simulations: I would be interested in seeing results from a simulation with a demography inferred from human data (e.g. Gutenkunst et al 2009 or Gravel et al 2011) to see the effect of recent growth and low levels of migration between populations on the power of VolcanoFinder. In addition, there are several data sets that have small sample sizes (e.g. SGDP) where it would be interesting to see the results of VolcanoFinder. I would like to see a simulation examining the effects of low sample size on the power of VolcanoFinder (how low can one go?).

3. Discussion: Several papers have reported on signals of adaptive introgression (e.g. BNC2 (Vernot and Akey 2014, Sankararaman et al 2014), OAS (Mendez et al 2013; Sams et al 2016), and several signals from Gittelman et al 2016, Browning et al 2018 & Durvasula and Sankararaman 2019) and as far as I can tell there aren't any overlaps with these studies. Many of those studies used an allele/haplotype frequency cutoff to infer adaptive introgression rather than the more sophisticated approach used here so I suspect there are many false positives in those lists. In addition, the strict filtering used by the authors here could have masked out some of the previously found signals (thereby removing some of the false positives). A discussion of the lack of overlap would be interesting to readers.

Minor comments:

Labels on fig 5,6 should be bigger. Explain the X axis

Table S3.1 appears to be cut off

Typos:

Page 18, line 317 "he mutation rate" should be "the mutation rate"

Page 25, line 451 "looses" should be "loses"

Page 32, line 574 "apolipoportein" should be "apolipoprotein"

Reviewer #2: Review of Setter, Mousset, et al.

In this manuscript the authors develop simple theoretical approximations to obtain the site frequency spectrum (SFS) from a model of adaptive introgression between species. With these approximations in hand, the authors then extend the sweepfinder machinery to look for adaptive introgression events using a composite likelihood estimator, named volcanofinder. The authors then benchmark the performance of volcanofinder and finally apply it to human data.

Generally I find this paper to be clearly written, quite timely, and the models are presented to be exceptionally clear. Most of my comments are aimed at improving the presentation even further, but I do have some additions that the authors should perform to strengthen the paper.

Major issues:

1) The authors are comparing the performance of volcanofinder to methods which are aimed at finding sweeps in single populations or balancing selection. While this is fine, the authors also need to compare volcanofinder to methods that are aimed at finding introgression writ large (with or without an adaptive sweep). I would suggest as a baseline S* from Jeff Wall or one of the newer supervised machine learning methods.

2) The authors also need to look at the robustness of volcanofinder to a few misspecified models that haven’t been looked at, namely a single sweep in the focal population without introgression, and background selection. Both of these additions should follow the section that is titled “Robustness to long term balancing selection”.

3) The “outlier” study design as presented is unconvincing. The authors are treating the 10^4 200kb chucks they have simulated as a single genome on which to perform outlier studies. This is inappropriate as in truth as outlier study is performed on a genome where all chromosomes share a pedigree and each chromosome itself is a single tree sequence. I would encourage the authors to either abandon this section, or more preferably, to do proper chromosome-scale simulations on which to base an outlier test. Related: having the “number of peaks” as the x-axis on Figs 5&6 is difficult to interpret, although I understand (I think) what the authors are trying to show. This should be unpacked a bit more in the text if kept.

4) Lines 682-686—the authors are making too light of the fact that “neutral admixture” severely limits the power of volcanofinder. Indeed *any* adaptive introgression will be accompanied by an even larger amount of non-beneficial introgression, so this is a more appropriate null background to be working in. The authors should recast the paper in this light as this is the biologically meaningful scenario, not finding an introgressed region in a genome that has not encountered introgression.

Minor issues:

1) Line 45—the last sentence of this paragraph is quite cryptic. How could recurrent hybridization lead to a soft sweep exactly?

2) Line 100—at this point the authors should explain what they mean by “complete lineage sorting in the ancestor” being assumed.

3) Line 141- “rd” should be defined here and its units made clear

4) Lines 208-214. This section is unclear to this reader. Which this scaling issue is important the authors are not helping the reader to follow what is happening.

5) Line 344—this section on the SFS should be moved either to the supplement, or directly after the model is introduced. This is a strange bit to have here.

6) Line 346—should say the sample size clearly here

7) Line 382—unclear at this point in the text what is meant by the “95% probability” of a sweep. This should be made clearer. Also can’t the authors just condition on the sweep having occurred?

8) Lines 432-438. A simpler way to present this information would be to just give the AUCs for the ROC curves.

9) Lines 461-468—again wondering why the authors just don’t throw out the simulations without a sweep. It would make this section of the paper much easier to the naïve reader.

10) Lines 545-547—It would be nice if the overlap or lack thereof in the manhattan plots were quantified. What percentage are the same? What is the expectation under independence?

11) Line 556 and following—this is not and McDonald-Kreitman test per se. It is a 2x2 contingency test of polymorphism and divergence however.

12) Line 588—it is unclear what the authors mean when they say that the a region does not “exhibit high CLR scores despite the region devoid of data”. If all missing data causes high CLR scores shouldn’t the authors simply adjust the output with a heuristic that says there is too little data here to calculate a CLR?

13) Line 733—this is a strange collection of papers to be citing for using ML for finding introgression

14) Line 950—all code should be deposited on a public repository. Software being “available upon request” is not acceptable in 2019.

Reviewer #3: This study was motivated by an increasing number of adaptive evolution discovered to be driven by positive selection on an introgressed variant. They found analytic approximation for the volcano pattern of polymorphism and turned it into a statistical method for genome scan. I found this study very timely and rigorous. I do not have any major point for criticism. Although this manuscript might be unnecessarily long, I think it is ready to be published after revisions to address the following minor issues.

My comments:

1. Lines 239-247. It will be nice if it is mentioned which specific panels in Figure 3 this section is talking about. In addition, “all B lineages” (line 241) might be better changed to “all B-linked lineages”, because whether a given lineage is B or b seemed to be defined in term of the final state shown in Table 1.

2. It was initially confusing to follow the derivation of SFS on page 16 because, I think, the assumption of infinite site model (on the entire genealogy linking recipient and outgroup sequences only one mutation event can be mapped) and exclusion of sites that are invariant over both recipient and outgroup was not emphasized enough. Only under that assumption, S_0(n) is understood as the probability of derived allele on the outgroup only.

3. line 317, he -> the.

4. line 374. I believe it is Text S2.4, not S1.3.

5. line 430. How “peaks” are defined is an important issue and should be briefly mentioned in Result. In Methods, the definition is not consistent: it is either a separation by less than 10 LR values (line 1053) or a fixed value of 15kb as minimum distance between peaks (line 1080). I wonder whether a better (logical) way of merging sites of significant LRs into a peak can be devised. For example, the minimum distance between peaks might be given proportional to the estimated strength of selection (s^).

6. lines 490-502. The effect of varying window size for a given Ns was not shown, not in Fig. S2.7.

7. line 542. Why non-synonymous differences only? Isn’t it more informative to use both synonymous and non-synonymous differences in finding introgression candidates?

8. line 556 and others. I think it is more appropriate to call it HKA (Hudson-Kreitman-Aguade) test rather than MK test.

9. line 566. The top rows of Table S3.1 are invisible in the manuscript file.

10. line 581. Apolipoprotein

11. It is difficult to follow lines 586-591. Fig S3.3A -> Fig 3.3F (?). What does it mean by “despite the region devoid of data” in CEU?

12. I think Discussion and conclusion can be shortened. Throughout the manuscript, similar information is given repetitively. For example, lines 1043-1049.

13. Maybe a direction of further development, such as detection of incomplete sweep of introgressed variant, can be mentioned?

**Have all data underlying the figures and results presented in the manuscript been provided?**

Reviewer #1: Yes

Reviewer #2: Yes

Reviewer #3: Yes

PLOS authors have the option to publish the peer review history of their article (what does this mean?). If published, this will include your full peer review and any attached files.

Reviewer #1: No

Reviewer #2: No

Reviewer #3: No

---

## [Decision Letter · Decision Letter 1]

18 May 2020

Dear Dr Setter,

We are pleased to inform you that your manuscript entitled "VolcanoFinder: genomic scans for adaptive introgression" has been editorially accepted for publication in PLOS Genetics. Congratulations!

Yours sincerely,

Alex Buerkle

Associate Editor

PLOS Genetics

Bret Payseur

Section Editor: Evolution

PLOS Genetics

Comments from the reviewers (if applicable):

Both reviewers are pleased with the revisions to this manuscript and are strongly supportive of its publication. I appreciate the authors' attention to the comments and suggestions from the first round of review.

Reviewer's Responses to Questions

**Comments to the Authors:**

Reviewer #1: The authors have done a nice job of responding to my comments and an impressive amount of work. I think this method will be very valuable to the community and am looking forward to seeing it used.

Reviewer #3: This revised version of manuscript addressed all of my comments in the previous round of review. I do not have any more concerns and thus think this paper is ready to be published.

**Have all data underlying the figures and results presented in the manuscript been provided?**

Reviewer #1: Yes

Reviewer #3: Yes

PLOS authors have the option to publish the peer review history of their article (what does this mean?). If published, this will include your full peer review and any attached files.

Reviewer #1: No

Reviewer #3: Yes: Yuseob Kim

**Data Deposition**

http://datadryad.org/submit?journalID=pgenetics&manu=PGENETICS-D-19-01183R1

**Press Queries**

---

## [Editor Report · Acceptance letter]

11 Jun 2020

PGENETICS-D-19-01183R1 

VolcanoFinder: genomic scans for adaptive introgression 

Dear Dr Setter, 

We are pleased to inform you that your manuscript entitled "VolcanoFinder: genomic scans for adaptive introgression" has been formally accepted for publication in PLOS Genetics! Your manuscript is now with our production department and you will be notified of the publication date in due course.

With kind regards,

Matt Lyles

PLOS Genetics

On behalf of:
